# Image Super-Resolution via Latent Diffusion: A Sampling-Space Mixture of Experts and Frequency-Augmented Decoder Approach

## Abstract

The recent use of diffusion prior, enhanced by pre-trained text-image models, has markedly elevated the performance of image super-resolution (SR). To alleviate the huge computational cost required by pixel-based diffusion SR, latent-based methods utilize a feature encoder to transform the image and then implement the SR image generation in a compact latent space. Nevertheless, there are two major issues that limit the performance of latent-based diffusion. First, the compression of latent space usually causes reconstruction distortion. Second, huge computational cost constrains the parameter scale of the diffusion model. To counteract these issues, we first propose a frequency compensation module that enhances the frequency components from latent space to pixel space. The reconstruction distortion (especially for high-frequency information) can be significantly decreased. Then, we propose to use Sample-Space Mixture of Experts (SS-MoE) to achieve more powerful latent-based SR, which steadily improves the capacity of the model without a significant increase in inference costs. These carefully crafted designs contribute to performance improvements in largely explored $4\times$ blind super-resolution benchmarks and extend to large magnification factors, i.e., $8\times$ image SR benchmarks.

## 1 Introduction

Diffusion models have quickly emerged as a powerful class of generative models, pushing the boundary of text-to-image generation, image editing, text-to-video generation, and more visual tasks (Sohl-Dickstein et al., 2015; Song & Ermon, 2019; Song et al., 2020; Ho et al., 2020). In this paper, we explore the potential of diffusion models to tackle the long-standing and challenging image super-resolution (SR) task.

Let us revisit the diffusion model in the context of generation space. Early diffusion models, operating in the high-dimensional pixel space of RGB images, demand substantial computational resources. To mitigate this, the Latent Diffusion Model (LDM) (Rombach et al., 2021) uses VQGAN to shift the diffusion process to a lower-dimensional latent space, maintaining generation quality while reducing training and sampling costs. Stable Diffusion further enhances LDM (Rombach et al., 2021) by scaling up the model and data, creating a potent text-to-image generator that has garnered significant attention in the generative AI field since its release.

However, a significant challenge arises when dealing with higher compression rates, drastically affecting detail consistency. As noted in studies (Kim et al., 2020; Rahaman et al., 2018), the convolutional nature of autoencoders tends to favor learning low-frequency features due to spectral bias. So the escalation in compression rate leads to loss of visual signals in the high-frequency spectrum, which embodies the details in pixel space. While some image synthesis researches (Lin et al., 2023b; Zhu et al., 2023) have recognized and addressed these issues, they have received little attention in the field of super-resolution(Wang et al., 2022a; Chung et al., 2022b; Lin et al., 2023a). StableSR(Wang et al., 2023) is one of the few models that fine-tune the autoencoder decoder with the CFW module, offering a potential solution to this problem in the spatial domain.

Regarding the training of the diffusion-based SR model, one approach(Wang et al., 2022b; Chung et al., 2022a; Kawar et al., 2022) involves using the pre-trained Stable Diffusion model, incorporat-

ing certain constraints to ensure fidelity and authenticity. However, the design of these constraints presupposes knowledge of the image degradations, which are typically unknown and complex. As a result, these methods often demonstrate limited generalizability. Another approach to address the above challenge involves training a super-resolution (SR) model from scratch, as seen in studies (Saharia et al., 2021; Li et al., 2021; Rombach et al., 2021; Sahak et al., 2023). To maintain fidelity, these methods use the low-resolution (LR) image as an additional input to limit the output space. Although these approaches have achieved significant success, they often require substantial computational resources to train the diffusion model, especially when the dimension exceeds $512 \times 512$ or $1024 \times 1024$. In this context, the super-resolution models are relatively small, possessing fewer parameters than the image generation model.

To this end, we aim to enhance the diffusion model for SR by fixing the decoding distortion and enlarging the diffusion model capacity without significantly increasing computational cost. We first propose a frequency compensation module that enhances the frequency components from latent to pixel space. The reconstruction distortion can be significantly decreased by better aligning the frequency spectrums of the high-resolution and reconstructed images. Then, we propose to use the Sample-Space Mixture of Experts (SS-MoE) to achieve stronger latent-based SR, which steadily improves the capacity of the model in an efficient way. Our approach allows for enlarging the model size without incurring significant computational costs during training and inference.

In summary, we highlight our contributions in three aspects:

- We identify the issue of information loss within the latent diffusion model used for image SR. In response, we propose a frequency-compensated decoder complemented by a refinement network. This innovative approach is designed to infuse more high-frequency details into the reconstructed images, thereby enhancing the overall image quality.

- We design sampling-space MoE to enlarge the diffusion model for image SR. This allows for enhanced high-resolution image processing without necessitating a substantial increase in training and inference resources, resulting in optimized efficiency.

- We evaluate the model on $4\times$ Blind SR and $8\times$ Non-Blind SR benchmarks, employing both quantitative and qualitative assessment methods. Additionally, we conduct essential ablation studies to validate the design choices of the models. Experiment results show that we achieved solid improvement in terms of perceptual quality, especially in $8\times$ SR.

## 2 RELATED WORK

**Image SR.** Image SR aims to restore an HR image from its degraded LR observation. Recent advancements (Liu et al., 2022) in Blind Super-Resolution (BSR) have delved into more intricate degradation models to mimic real-world deterioration. Specifically, BSRGAN (Zhang et al., 2021b) is designed to emulate more realistic degradations using a random shuffling approach, while Real-ESRGAN (Wang et al., 2021) leverages "high-order" degradation modeling. Both methodologies employ Generative Adversarial Networks (GANs) (Goodfellow et al., 2014; Miyato et al., 2018) to understand the image reconstruction process amidst complex degradations. FeMaSR (Chen et al., 2022b) interprets SR as a feature-matching issue, utilizing the pre-trained VQ-GAN (Esser et al., 2020). Despite the utility of BSR techniques in mitigating real-world degradations, they fall short of generating realistic details.

**Diffusion Model for Image SR.** Using diffusion models in image SR signifies a burgeoning trend. The primary driving force behind this methodology is the exceptional generative capacity of diffusion models. Numerous research efforts have explored their use in image restoration tasks, specifically enhancing texture recovery (Ramesh et al., 2022; Rombach et al., 2021; Nichol et al., 2021). The remarkable generative prowess of these pre-trained diffusion models has been showcased, underscoring the critical need for high-fidelity inherent in SR. Based on the training strategy, these studies can be broadly classified into two categories: supervised training and zero-shot methods. The first category (Saharia et al., 2021; Li et al., 2021; Niu et al., 2023; Sahak et al., 2023) is committed to optimizing the diffusion model for SR from the ground up through supervised learning. The zero-shot approach (Choi et al., 2021; Wang et al., 2022a; Chung et al., 2022b; Fei et al., 2023) aims to leverage the generative priors in the pre-trained diffusion models for SR, by imposing certain constraints to ensure image fidelity. Those zeros-shot approaches usually show limited ability

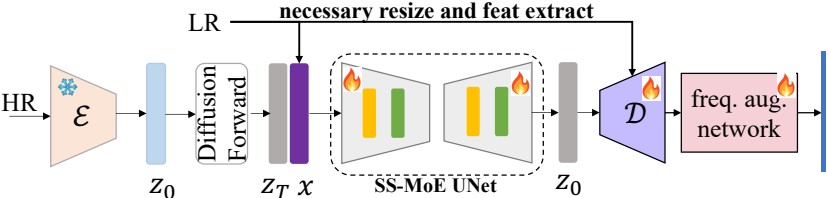

Figure 1: Latent diffusion model for image SR with SS-MoE and frequency augmented decoder.

to super-resolve and the supervised methods are constrained to limited model scale due to huge computation costs.

**Autoencoder in Diffusion Model.** To reduce the training and sampling costs linked to the diffusion model, StableDiffusion (Rombach et al., 2021) spearheads the use of DM-based generation in the latent space. This is achieved specifically through a pre-training phase for an autoencoder model (Esser et al., 2020), defined by an encoder-decoder architecture, to navigate the perceptual space proficiently. However, the high compression rate in the latent space often results in image distortion during the reconstruction of images from the low-dimensional latent space Lin et al. (2023b); Zhu et al. (2023). The lossy latent necessitates a more robust decoder to offset the information loss.

## 3 METHODOLOGY

### 3.1 PRELIMINARIES

Given a dataset of low-resolution and target image pairs, denoted as $\mathcal{D} = \{\boldsymbol{x}_i, \boldsymbol{y}_i\}_{i=1}^N$ drawn from an unknown distribution $p(\boldsymbol{x}, \boldsymbol{y})$. Image SR is a process of conditional distribution modeling $p(\boldsymbol{y}|\boldsymbol{x})$, and it is a one-to-many mapping in which many target images may be consistent with a single low-resolution image. Our objective is to learn a parametric approximation to $p(\boldsymbol{y}|\boldsymbol{x})$ through a stochastic iterative refinement process that transforms a source image $\boldsymbol{x}$ into a target image $\boldsymbol{y}$. We tackle this problem by adapting the diffusion probabilistic (DDPM) model(Ho et al., 2020; Sohl-Dickstein et al., 2015) to conditional image SR.

DDPM is the first diffusion-based method introduced in (Sohl-Dickstein et al., 2015), which consists of a diffusion process and a denoising process. In the diffusion process, it gradually adds random noises to the data $x$ via a T-step Markov chain (Kong & Ping, 2021). The noised latent variable at step t can be expressed as:

$$\mathbf{z}_t = \sqrt{\hat{\alpha}_t}\boldsymbol{y} + \sqrt{1 - \hat{\alpha}_t}\epsilon_{\mathbf{t}}, \text{with } \hat{\alpha}_t = \prod_{k=1}^t \alpha_k \quad \epsilon_{\mathbf{t}} \sim \mathcal{N}(\mathbf{0}, \mathbf{1}), \tag{1}$$

where $\alpha_t \in (0, 1)$ is the corresponding coefficient. For a $T$ that is large enough, e.g., $T = 1000$, we have $\sqrt{\hat{\alpha}_T} \approx 0$ and $\sqrt{1 - \hat{\alpha}_T} \approx 1$. And $\mathbf{z}_T$ approximates a random Gaussian noise. Then, the generation of $\mathbf{x}$ can be modeled as iterative denoising.

Ho et al. (2020) connect DDPM with denoising score matching and propose a $\epsilon-$prediction form for the denoising process:

$$\mathcal{L}_t = \|\epsilon_t - f_\theta(\mathbf{z}_t, \boldsymbol{x}, t)\|^2, \tag{2}$$

where $f_\theta$ is a denoising neural network parameterized by $\theta$, and $\mathcal{L}_t$ is the training loss function. The cornerstone of this design is the denoising neural network, which is typically a UNet.

During inference, we reverse the diffusion process through iterative refinement, taking the form of:

$$\boldsymbol{y}_{t-1} \leftarrow \frac{1}{\sqrt{\alpha_t}}\left(\boldsymbol{y}_t - \frac{1 - \alpha_t}{\sqrt{1 - \gamma_t}}f_\theta(\boldsymbol{x}, \boldsymbol{y}_t, \gamma_t)\right) + \sqrt{1 - \alpha_t}\epsilon_t, \quad \boldsymbol{y}_T \sim \mathcal{N}(\mathbf{0}, \boldsymbol{I}). \tag{3}$$

The proposed latent diffusion model for image SR is illustrated in Fig. 1. It consists of multiple SS-MoE UNet and a frequency-compensated autoencoder that will be illustrated in 3.2 and 3.3.

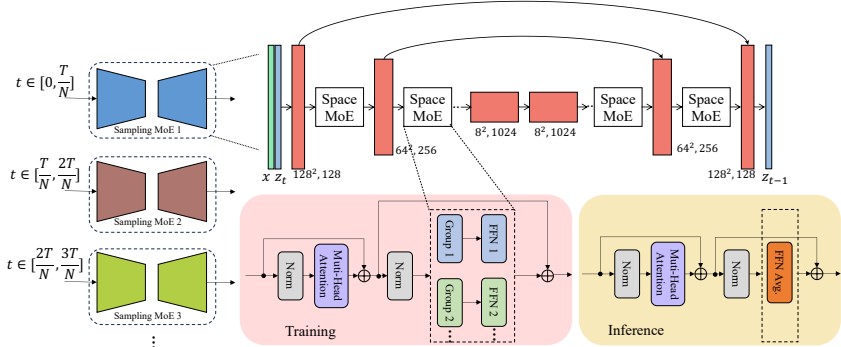

Figure 2: Sampling-Space MoE of the denoising UNet for image SR. "FFN Avg." means averaging the weights of all experts into one FFN for inference after training.

## 3.2 DENOISING UNET WITH SS-MOES

The denoising UNet structure, denoted as $f_\theta$, is inspired by the Latent Diffusion Model (LDM) (Rombach et al., 2021), and it incorporates residual and self-attention blocks as its core building elements. To make the model conditional on the input $\boldsymbol{x}$, we employ bicubic interpolation (Saharia et al., 2021) to up-sample the low-resolution image to match the target resolution. The up-sampled result is concatenated with $\boldsymbol{z}_t$ along the channel dimension, see Fig. 2.

**Sampling MoE.** The quality of the image can be significantly improved by utilizing a time-mixture-of-experts (time-MoE) method, a concept derived from earlier studies (Xue et al., 2023; Feng et al., 2022; Balaji et al., 2022). Similarly, diffusion-based SR is also a diffusion process that progressively introduces Gaussian noise to an image over a sequence of timesteps, $t = 1, ..., T$. The image generator is trained to reverse this process given an upsampled low-resolution image as the condition, denoising the images from $t = T$ to $t = 1$. Each timestep is designed to denoise a noisy image, gradually converting it into a clear high-resolution image. It is important to note that the complexity of these denoising steps fluctuates based on the level of noise present in the image. For instance, when $t = T$, the input image $\boldsymbol{x}_t$ for the denoising network is heavily noisy. However, when $t = 1$, the image $\boldsymbol{x}_t$ is much closer to the original, aka less noisy image. So we divide all timesteps uniformly into $N$ stages consisting of consecutive timesteps and assign a single Sampling Expert to one stage. Since only a single expert network is activated at each step, the scale and capacity of our model can expand with computational overhead remaining the same during inference, regardless of an increase in the number of experts. We use $N = 4$ to assure that all experts can be loaded on a GPU when inference.

**Space MoE.** We create MoE layers with $N$ spatial experts (i.e., $N$ FFNs) $\{E_1, E_2, ...E_N\}$ after existing multi-head attention to scale the denoising UNet. For a batch of input tokens $(B, L, d)$, where $L = hw$, $B$ denotes the batch size and $h, w, d$ denote the height, width and channel number of a feature map respectively. Assuming $L$ is divisible by $N$, we randomly split the tokens into $N$ groups and then processed with experts:

$$\{x_1, x_2, ..., x_L\} \xrightarrow{\text{group split}} \{X_1, X_2, ..., X_N\}, \; y = E_i(x). \tag{4}$$

Given the weights if $N$ experts $\{W_1, W_2, ..., W_N\}$, weight sharing is performed among all experts during training:

$$\overline{W_i} = \gamma W_i + (1 - \gamma)\overline{W}_j \text{ with } \overline{W}_j = \sum_{j \neq i}^{N} \frac{1}{N - 1} W_j, \tag{5}$$

where $\overline{W_i}$ denotes the updated weight of the $i$-th expert. Conceptually, we update the weight of each expert by averaging the weights of the other experts. The momentum coefficient $\gamma \in [0, 1)$ regulates the degree of information exchange among the experts. The momentum update, as shown in Eq. 5, ensures a smoother evolution of each expert. Each expert carries a substantial dropout (i.e., $\frac{N-1}{N}$) and they collectively evolve through momentum updates. A relatively large momentum (e.g., $\gamma = 0.999$) works better than a smaller value (e.g., $\gamma = 0.9$), suggesting that smaller $\gamma$ could probably lead to weight collapse (identical weights across all experts).

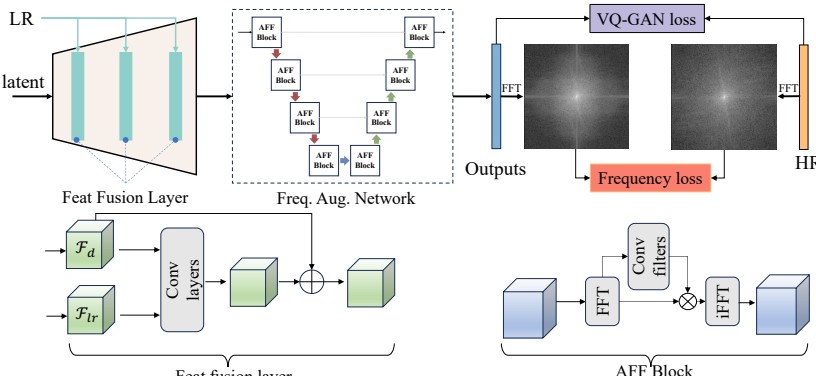

Figure 3: The proposed frequency-augmented decoder is conditioned on the low-resolution image and uses the AFF block (Huang et al., 2023) to reduce information loss in the frequency domain.

After training, each space MoE layer is converted into an FFN layer by simply averaging the experts: $\text{FFN} = \frac{1}{N}\Sigma_{i=1}^{N}E_i$. In this way, introducing space MoE to augment the denoising UNet only incurs the computation overhead of a single FFN.

## 3.3 Frequency Compensated Decoder

To address the information loss using autoencoder, we propose improving the image super-resolution quality by augmenting the decoder with a frequency-compensated loss and network. To be specific, the frequency-augmented decoder comprises a VQGAN(Esser et al., 2020) decoder conditioning on low-resolution inputs similar to (Wang et al., 2023) and a refinement network utilizing frequency operators along with a frequency loss for optimization, as depicted in Fig. 3.

**LR-conditioned Decoder.** Conditioning on low-resolution inputs has been proven to enhance the reconstruction fidelity for image SR (Wang et al., 2023; 2018a). We add a feature extractor to get the LR image representation for conditions during decoding. Since only several encoding features are needed, some layers like VQGAN's middle blocks can be dropped to save memory and computation cost during inference. The fusion of LR features $F_{lr}$ and decode latent $F_d$ can be formulated as $F_m = F_d + \mathcal{C}(F_{lr}, F_d; \theta)$, where $\mathcal{C}(\cdot; \theta)$ is a sequences of trainable convolution layers as designed in (Wang et al., 2023).

**Refinement Network.** We further use a tiny UNet model with frequency augmentation operation to address information loss. It is inserted after the last upsample block of the VQGAN decoder. The UNet model consists of six AFF blocks proposed by (Huang et al., 2023), ie. sequences of activations, linear layers, and adaptive frequency filters. The frequency operator first transforms the input latent into the frequency domain using the Fourier transform and then applies semantic-adaptive frequency filtering through element-wise multiplication.

To optimize the frequency-augmented decoder, we use frequency loss $\mathcal{L}_{\text{freq}}$ (Jiang et al., 2020) in addition to VQ-GAN loss $\mathcal{L}_{\text{VQ-GAN}}$ (Esser et al., 2020)for reconstruction:

$$\mathcal{L} = \mathcal{L}_{\text{VQ-GAN}} + \lambda\mathcal{L}_{\text{freq}} \text{ , with, } \mathcal{L}_{\text{freq}} = \frac{1}{MN}\sum_{u=0}^{M-1}\sum_{v=0}^{N-1} w(u,v)\left|F_r(u,v) - F_f(u,v)\right|^2. \quad (6)$$

We set $\lambda = 10$ by default; the matrix element $w(u,v)$ is the weight for the spatial frequency at $(u,v)$; $F_r(u,v), F_f(u,v)$ are the FFT results of ground-truth and reconstruction images.

## 4 Experiments

**Datasets.** We train and test our method on $4\times$ and $8\times$ super-resolution with synthesized and real-world degradation settings. For each task, there are two training stages, stage 1 for Sampling-Space MoE and stage 2 for Frequency Compensated Decoder. In the first stage, degraded pipelines are

Table 1: Quantitative Results on 4× SR benchmarks. The 1st and the 2nd best performances are highlighted in red and blue, respectively. † means reproducing with the official model.

| Datasets | Metrics | RealSR | BSRGAN | Real-ESRGAN+ | DASR | FeMaSR | LDM | StableSR | StableSR† | OURS |
|---|---|---|---|---|---|---|---|---|---|---|
| DIV2K Valid | PSNR↑ | 22.36 | 22.71 | 22.93 | 22.70 | 21.73 | 21.86 | 23.26 | 21.81 | 22.11 |
| | SSIM↑ | 0.5559 | 0.5911 | 0.6144 | 0.5988 | 0.5692 | 0.5554 | 0.5726 | 0.5534 | 0.5775 |
| | LPIPS↓ | 0.6191 | 0.3546 | 0.3229 | 0.3545 | 0.3389 | 0.3264 | 0.3114 | 0.3143 | 0.2821 |
| | FID↓ | 71.85 | 49.0 | 40.49 | 51.79 | 41.26 | 27.36 | 24.44 | 25.64 | 25.49 |
| | MUSIQ↑ | 27.20 | 61.20 | 60.70 | 57.26 | 57.83 | 62.8906 | 65.92 | 66.78 | 64.78 |
| | NIQE↓ | 7.71 | 4.84 | 4.88 | 4.95 | 4.95 | 5.64 | - | 4.81 | 4.72 |
| RealSR | PSNR↑ | 27.77 | 26.55 | 25.82 | 27.06 | 25.42 | 25.46 | 24.65 | 25.19 | 24.68 |
| | SSIM↑ | 0.7760 | 0.7742 | 0.7700 | 0.7833 | 0.7458 | 0.7145 | 0.7080 | 0.7227 | 0.7352 |
| | LPIPS↓ | 0.3805 | 0.2623 | 0.2670 | 0.2923 | 0.2855 | 0.3159 | 0.3002 | 0.2974 | 0.2719 |
| | FID↓ | 88.26 | 97.09 | 93.74 | 92.82 | 95.32 | 83.98 | - | 80.25 | 83.59 |
| | MUSIQ↑ | 29.88 | 62.63 | 61.54 | 45.89 | 58.92 | 58.90 | 65.88 | 63.44 | 57.10 |
| | NIQE↓ | 8.14 | 5.87 | 6.15 | 6.48 | 5.90 | 6.78 | - | 6.35 | 5.96 |
| DRealSR | PSNR↑ | 31.73 | 29.98 | 29.69 | 31.14 | 27.54 | 27.88 | 28.03 | 29.00 | 29.35 |
| | SSIM↑ | 0.8563 | 0.8240 | 0.8277 | 0.8493 | 0.7638 | 0.7448 | 0.7536 | 0.7658 | 0.7946 |
| | LPIPS↓ | 0.3634 | 0.2757 | 0.2663 | 0.2873 | 0.3273 | 0.3379 | 0.3284 | 0.3353 | 0.3017 |
| | MUSIQ↑ | 23.91 | 55.12 | 52.21 | 40.50 | 52.56 | 53.72 | 58.51 | 56.72 | 42.32 |
| | NIQE↓ | 9.47 | 6.70 | 7.04 | 7.84 | 6.22 | 7.37 | - | 6.89 | 6.89 |

different for each task. For the 4× super-resolution, following StableSR (Wang et al., 2023), we combine images in DIV2K (Agustsson & Timofte, 2017), Flickr2K(Timofte et al., 2017) and OutdoorSceneTraining(Wang et al., 2018b) datasets as the training set. We additionally add the openImage dataset (Kuznetsova et al., 2020) for general cases. LR-HR pairs on DIV2K are synthesized with the degradation pipeline of Real-ESRGAN (Wang et al., 2021). The sizes of LR and HR patch are 128×128 and 512×512. For the 8× super-resolution, we only use DIV2K, Flickr2K and openImage dataset for training. LR images are with a size of 64×64 and obtained via default setting (bicubic interpolation) of Matlab function imresize with scale factor 8. In stage 2 of Frequency Compensated Decoder training, we adopt Sampling-Space MoE to generate $100k$ LR-Latent pairs for 4× and 8× SR given the LR images as conditions.

**Training.** We train all of our Sampling-Space MoEs for $100k$ steps with a batch size of 144. Moreover, training steps for Frequency Compensated Decoder is $50k$ and the batch size is 32. Following LDM(Esser et al., 2020), we use Adam optimizer, and the learning rate is fixed to $5 \times 10^{-5}$ and $1 \times 10^{-4}$ for SS-MOEs and FCD. All trainings are conducted on 8 NVIDIA Tesla 32G-V100 GPUs.

**Inference.** Consistent with stableSR, we implement DDPM sampling with 200 timesteps. However, fewer steps can yield comparable results, as discussed in Sec.4.3. We employ evaluation metrics including LPIPS (Zhang et al., 2018), FID (Heusel et al., 2017), MUSIQ (Ke et al., 2021) and NIQE (Mittal et al., 2012). PSNR and SSIM scores are also reported on the luminance channel in the YCbCr color space.

## 4.1 BENCHMARK RESULTS OF 4× BLIND IMAGE SR.

We first evaluate our method on blind super-resolution. For synthetic data, we follow the degradation pipeline of Real-ESRGAN(Wang et al., 2021) and generate $3k$ LR-HR pairs from DIV2K validation set. We compare our method quantitatively with GAN-based methods such as RealSR(Ji et al., 2020), BSRGAN(Zhang et al., 2021a), Real-ESRGAN+(Wang et al., 2021), DASR(Liang et al., 2022), FeMaSR(Chen et al., 2022a) and diffusion-based methods like LDM(Rombach et al., 2021) and StableSR. The quantitative results are shown in Tab. 1. Note that due to differences in making test sets, we reproduce StableSR using its official model and code. We can see that our approach outperforms state-of-the-art SR methods on perceptual metrics (including LPIPS, FID, and NIQE) and gets the best PSNR and SSIM among diffusion-based methods. Specifically, synthetic benchmark DIV2K Valid, our method achieves a 0.2821 LPIPS score, which is 10.24% lower than StableSR and at least 12.64% lower than other GAN-based methods. Besides, our method achieves the lowest LPIPS score among diffusion-based methods on the two real-world benchmarks (Cai et al., 2019; Wei et al., 2020), which clearly demonstrates the superiority of our approach. Note that although GAN-based methods like BSRGAN and Real-ESRGAN+ achieve good MUSIQ and NIQE scores, but fail to restore faithful details, such as textures and small objects, and generate blurry results as shown in Fig. 4. Compared with the diffusion-based methods, our method also produces more vi-

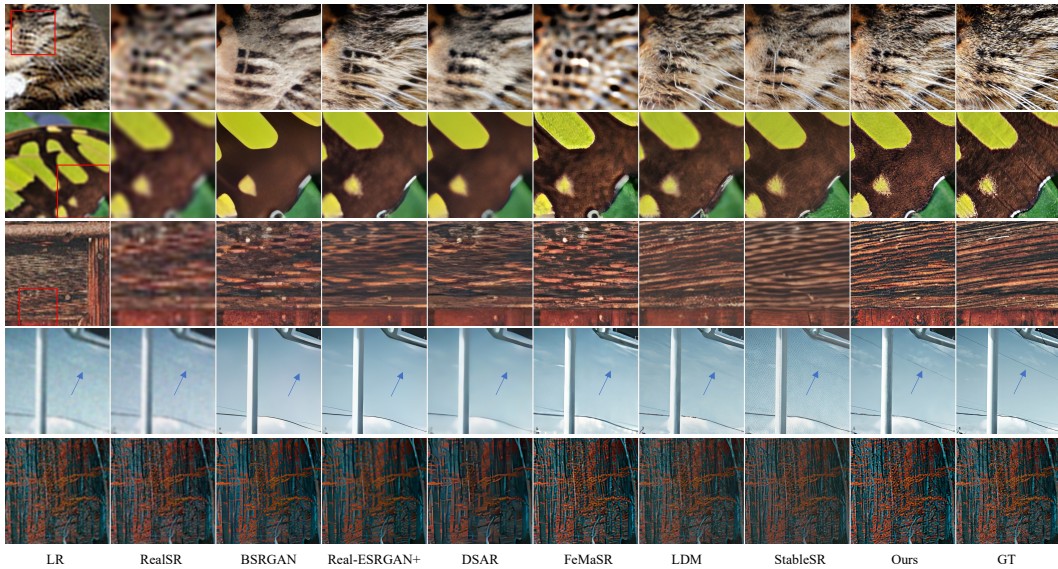

|     |     |     |     |     |     |     |     |     |     |
| --- | --- | --- | --- | --- | --- | --- | --- | --- | --- |
| LR | RealSR | BSRGAN | Real-ESRGAN+ | DSAR | FeMaSR | LDM | StableSR | Ours | GT |

Figure 4: Qualitative comparisons on $4\times$ SR ($128 \to 512$). Our method is capable of achieving better detail consistency and generating realistic texture details. (Zoom in for the best view)

sually promising results by reserving more high-frequency information and achieving better detail consistency.

## 4.2 BENCHMARK RESULTS OF $8\times$ NON-BLIND IMAGE SR.

We further validate the effectiveness of our method on $8\times$ SR. For the test set, we generate 660 LR-HR pairs from DIV2K validation set via bicubic interpolation with the scale factor 8. We compare to other state-of-the-art models that span from regression models having powerful architectures and/or generative formulations: RRDB(Wang et al., 2018c), ESRGAN(Wang et al., 2018c), SR-FLOW(Lugmayr et al., 2020), FxSR-PD(Park et al., 2022), LDM(Rombach et al., 2021). We use pre-trained models provided by the authors while for the non-provided $8\times$ SR model (RRDB and ESRGAN), we get unofficial released models from the github[1] of SRFLow. All results are tested on the same dataset using the official inference code. As Tab. 2 shows, our approach significantly outperforms competing methods on LPIPS, FID, and MUSIQ, and achieves top-2 in terms of NIQE. The qualitative results in Fig. 5 agree with the conclusions of the numerical results. It can be seen that our method can generate sharp images with high fidelity more naturally, while others tend to distort the characters or produce artifacts. Besides, our method can also generate realistic texture details, whereas others produce over-smooth results. Benefiting from the capacity and scalability of SS-MoE, our method has much more clear results compared to LDM's blurry output.

Table 2: Quantitative Results on synthetic $8\times$ SR benchmarks.

| Datasets | Metrics | Bicubic | RRDB | ESRGAN | SRFlow | FxSR-PD | LDM | OURS |
| --- | --- | --- | --- | --- | --- | --- | --- | --- |
| DIV2K Valid | PSNR↑ | 25.37 | 27.18 | 24.13 | 24.88 | 25.24 | 23.81 | 24.45 |
|  | SSIM↑ | 0.6361 | 0.6995 | 0.6035 | 0.6008 | 0.6312 | 0.5875 | 0.6142 |
|  | LPIPS↓ | 0.6055 | 0.4332 | 0.2767 | 0.2706 | 0.2425 | 0.3087 | 0.2321 |
|  | FID↓ | 118.63 | 92.4 | 58.64 | 59.52 | 55.0 | 63.07 | 44.49 |
|  | MUSIQ↑ | 20.80 | 46.43 | 55.84 | 55.20 | 62.10 | 63.82 | 64.17 |
|  | NIQE↓ | 11.31 | 8.65 | 3.97 | 4.60 | 5.24 | 5.75 | 4.34 |

---

[1]https://github.com/andreas128/SRFlow

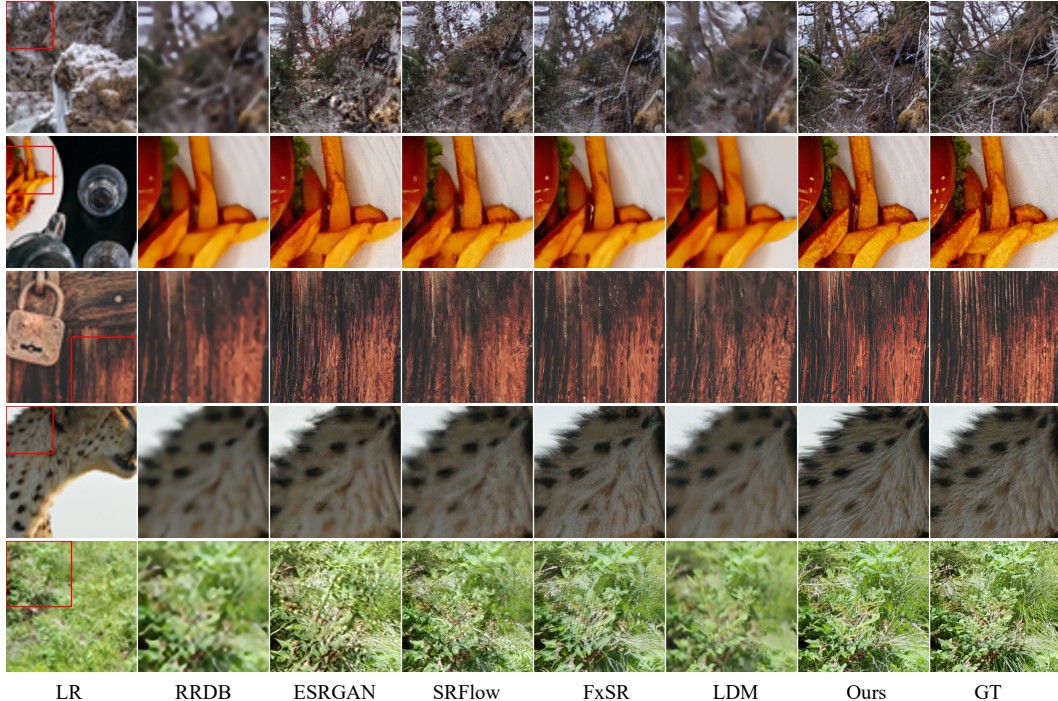

LR     RRDB    ESRGAN    SRFlow    FxSR    LDM    Ours    GT

Figure 5: Qualitative comparisons on 8× SR (64 → 512). Our method is capable of generating sharp images with high-fidelity texture details. (Zoom in for the best view)

Table 3: The effectiveness of SS-MoE evaluated on both 4× and 8×SR.

| Model | PSNR↑ | SSIM ↑ | LPIPS↓ | FID↓ | MUSIQ↑ |
|---|---|---|---|---|---|
| vanilla | 21.86/23.81 | 0.5554/0.5875 | 0.3264/0.3087 | 27.36/63.07 | 62.89/63.82 |
| w/o Sampling MoE | 22.09/24.94 | 0.5649/0.6176 | 0.3201/0.2497 | 26.21/46.34 | 63.14/63.32 |
| w/o Space MoE | 22.15/24.91 | 0.5680/0.6175 | 0.3134/0.2426 | 26.19/43.93 | 63.77/63.98 |
| w SS-MoE | 22.25/25.16 | 0.5725/0.6273 | 0.3031/0.2267 | 23.61/41.37 | 64.06/63.61 |

## 4.3 ABLATION STUDIES AND COMPUTATION COST ANALYSIS

**Ablation on SS-MoE.** We investigate the significance of our proposed Sampling-Space Mixture of Experts on both 4× and 8 × SR. Here, we use the original VAE to decode latents generated by different models. As shown in Tab. 3, the removals of Sampling MoE and Space MoE both lead to a noticeable performance drop in almost all evaluated metrics for different tasks, demonstrating both modules contribute to our approach's powerful generative ability and fidelity. Furthermore, we evaluate the models' performance under different sampling steps. As depicted in Tab. 4, for each sampling step Sampling-MoE can generate high-resolution images with better perceptual quality, showing its stronger denoise ability by modeling noises of different levels using multiple experts. We also notice that Sampling-MoE can achieve better FID and LPIPS with fewer steps. For example, the two metrics in both 4× and 8 × SR with $T = 50$ outperform Space-MoE model with $T = 200$, decreasing 75% sampling steps and resulting in more efficient diffusion-based SR.

**Ablation on FCD.** Then, we aim to illustrate the effectiveness of our proposed Frequency Compensated Decoder. The ablation experiments in Table. 5 have the same 25k training steps and are evaluated on 4× DIV2K. We use the VQ Model in LDM as the baseline

Table 5: Ablation Study of FCD.

| Model | PSNR↑ | SSIM↑ | LPIPS↓ | FID↓ | MUSIQ↑ | NIQE↓ |
|---|---|---|---|---|---|---|
| Baseline | 22.25 | 0.5725 | 0.3038 | 23.90 | 64.02 | 5.6093 |
| + AFF-Net | 22.16 | 0.5805 | 0.2808 | 24.52 | 63.66 | 4.4337 |
| + FFL loss | 22.01 | 0.5675 | 0.2892 | 24.94 | 63.64 | 4.5789 |
| + UNet + FFL | 22.19 | 0.5791 | 0.2856 | 25.09 | 63.94 | 4.3652 |
| + AFF + FFL | 22.23 | 0.5814 | 0.2815 | 24.30 | 64.03 | 4.3675 |

Table 4: Comparison SS-MOE and Space-Moe on DIV2K with different sampling steps.

| Task | Model | Sampling Step | PSNR↑ | SSIM ↑ | LPIPS ↓ | FID↓ | MUSIQ↑ |
|------|-------|---------------|-------|--------|---------|------|--------|
| 4×SR | Space-Moe | T=200 | 22.09 | 0.5649 | 0.3201 | 26.21 | 63.14 |
|      |           | T=100 | 22.26 | 0.5730 | 0.3188 | 26.25 | 63.15 |
|      |           | T=50  | 22.54 | 0.5852 | 0.3246 | 28.42 | 61.83 |
|      |           | T=20  | 23.05 | 0.6051 | 0.3547 | 36.08 | 57.39 |
|      | SS-MoE | T=200 | 22.24 | 0.5727 | 0.3031 | 23.61 | 64.06 |
|      |        | T=100 | 22.45 | 0.5824 | 0.3074 | 24.15 | 63.37 |
|      |        | T=50  | 22.70 | 0.5918 | 0.3125 | 25.87 | 62.12 |
|      |        | T=20  | 23.26 | 0.6141 | 0.3461 | 34.58 | 57.56 |
| 8×SR | Space-Moe | T=200 | 24.94 | 0.6176 | 0.2497 | 46.34 | 63.32 |
|      |           | T=100 | 25.16 | 0.6262 | 0.2495 | 45.99 | 62.64 |
|      |           | T=50  | 25.47 | 0.6381 | 0.2595 | 46.81 | 61.36 |
|      |           | T=20  | 26.07 | 0.6606 | 0.2829 | 51.91 | 57.49 |
|      | SS-MoE | T=200 | 25.16 | 0.6273 | 0.2267 | 41.37 | 63.61 |
|      |        | T=100 | 25.37 | 0.6365 | 0.2302 | 41.59 | 63.05 |
|      |        | T=50  | 25.60 | 0.6449 | 0.2361 | 42.68 | 62.06 |
|      |        | T=20  | 26.20 | 0.6676 | 0.2668 | 48.8 | 58.41 |

and add AFF-Net and FFL Loss progres-
sively. As shown in Table. 5, frequency refinement introduced by AFF Net and FFL loss improves the perceptual quality of images, with 7.3% improvement of LPIPS and 22.1% improvement of NIQE in contrast to baseline. Compared with UNet+FFL, AFF+FFL achieves lower LPIPS and FID, indicating better realism and suggesting the effectiveness of frequency operation.

**Parmeter and Computational Cost Analysis**
We further evaluate our method against other diffusion-based SR methods, including LDM and StableSR on 4× SR in terms of the parameter number and FLOPs. The results are shown in Tab. 6. We calculate FLOPs for one denoising step and a single SR inference separately and timesteps are set to 200 when inference. Notice that our model's parameter num-

Table 6: Parameter and computation cost comparison on 4× SR using 200 timesteps for inference.

| Method | Total Param (M) | UNet Param (M) | VAE Param (M) | FLOPs(T) per step | Total FLOPs(T) |
|--------|-----------------|----------------|---------------|-------------------|----------------|
| LDM | 168.95 | 168.95 | 168.95 | 0.1608 | 33.43 |
| Ours | 605.30 | 605.30 | 605.30 | 0.1658 | 35.47 |
| StableSR | 1409.11 | 1409.11 | 1409.11 | 0.4162 | 86.27 |

ber, 605.30, includes SS-MoE with four experts and a frequency-augmented decoder. Benefitting from SS-MoE, the parameter number of our method increases by 436.35M and the FLOPs only increase by 3.1% and 6.1% compared with LDM. As for StableSR, it utilizes the stable diffusion 2.1 architecture along with a half UNet, resulting in approximately 2.5 times FLOPs compared to our method, highlighting our lightweight nature.

## 5 CONCLUSION

Unlike existing pixel diffusion-based SR methods that require huge calculating resources, we have introduced a latent diffusion model for efficient SR. We propose Sampling-Space MoE to enlarge the diffusion model without necessitating a substantial increase in training and inference resources. Furthermore, to address the issue of information loss caused by the latent representation of the diffusion model, we propose a frequency-compensated decoder to refine the details of super-resolution images. Extensive experiments on both Blind and Non-Blind SR datasets have demonstrated the superiority of our proposed method.

**Limitations.** While our method has demonstrated promising results, the potential of diffusion-based methods has not been fully explored. We encourage further exploration in Latent Diffusion SR to achieve stronger generalizability in real-world SR. Increasing the size of model and using more degradation pipelines of data may help alleviate the problem. Our frequency-compensation decoder does not completely address the distortion caused by latent space compression. Expanding the latent feature channel might be a solution to further increase the reconstruction accuracy, but it will also result in a model that is more difficult to converge.

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

Table 7: Hyper-parameters and values in SS-MoE and Frequency Compensated Decoder.

| Configs/Hyper-parameters | Values |
|---|---|
| **SS-MoE** | |
| $f$ | 4 |
| $z$-shape | 35.47 |
| Channels | 160 |
| Channel multiplier | [1, 2, 4, 4] |
| Attention resolutions | [16, 8] |
| Head channels | 32 |
| Architectures of Space MoE | FFN |
| Activations in Space MoE | GELU |
| Number of Sampling MoEs | 4 |
| Time stage split | [(1000, 750), (750, 500], (500, 250], (250, 0]] |
| | |
| **Frequency Compensated Decoder** | |
| Embed dim | 4 |
| Number of embed | 8192 |
| Double z | False |
| Z channels | 3 |
| Channels | 128 |
| Channel multiplier | [1, 2, 4] |
| Number of Residual blocks | 2 |
| Attention resolutions | [ ] |
| Dropout rate | 0.0 |
| Number of feat fusion layers | 1 |
| Number of AFF blocks | 1 |

Table 8: Hyper-parameters and values in two stages' training.

| Configs/Hyper-parameters | Stage1 | Stage2 |
|---|---|---|
| Loss | L2 | L1, LPIPS, GAN Loss, FFL Loss |
| Training steps | 1e5 | 5e4 |
| Learning rate | 5e-5 | 1e-4 |
| Batch size per GPU | 9 | 1 |
| Accumulate grad batches | 2 | 4 |
| Number of GPU | 8 | 8 |
| GPU-type | V100-32GB | V100-32GB |

## A  IMPLEMENT DETAILS

In this part, we illustrate the details of our method, including model architecture and training setting. To be specific, the framework contains two parts, denoise UNet and Frequency Compensated Decoder, which correspond to the training stage1 and training stage2 respectively. The denoising UNet follows the architecture of Latent Diffusion Model and the Frequency Compensated Decoder is based on VQModel. All hyperparameters are as shown Tab. 7 and Tab. 8.

## B  USER STUDY

To further confirm the superiority of our method, we conduct a user study on 210 real-world LR images. These images were collected from three different datasets, namely DIV2K, RealSR, and DRealSR, with each dataset contributing 70 images. We compare our approach with 7 commonly used SR methods with competitive performance, i.e., RealSR, BSRGAN, Real-ESRGAN+, DASR, FeMaSR, LDM and StableSR. The comparison is conducted in pairs, i.e., given an LR image as reference, the subject is asked to choose the better HR image generated from either our method

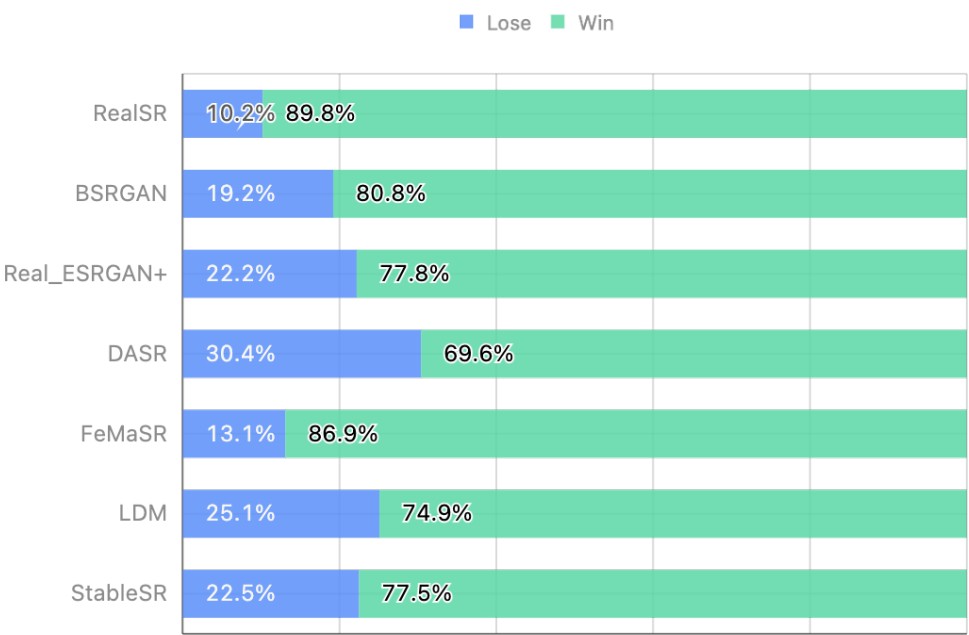

Figure 6: Ours vs. Prior Work in $4\times$ SR.

**★1**   Which ofthe two images is a better high quality version of the low resolution image in the left?"

Low−Resolution Image    Result1    Result2

◯ Result 1

◯ Result 2

Figure 7: an Example of User Study.

or others. The example of the user study is shown in Fig. 7. The LR images are divided into 30 pairs for each method. 17 users are recruited to conduct this user study under detailed instruction, so there are $30 \times 7 \times 17$ votes in total. The win rates of our method are shown in Fig. 7. It can be observed that our method outperforms all 7 competitive methods by a large margin, consistently gaining approximately 70% of the votes all the time, indicating the substantial superiority of the proposed method on human perception.

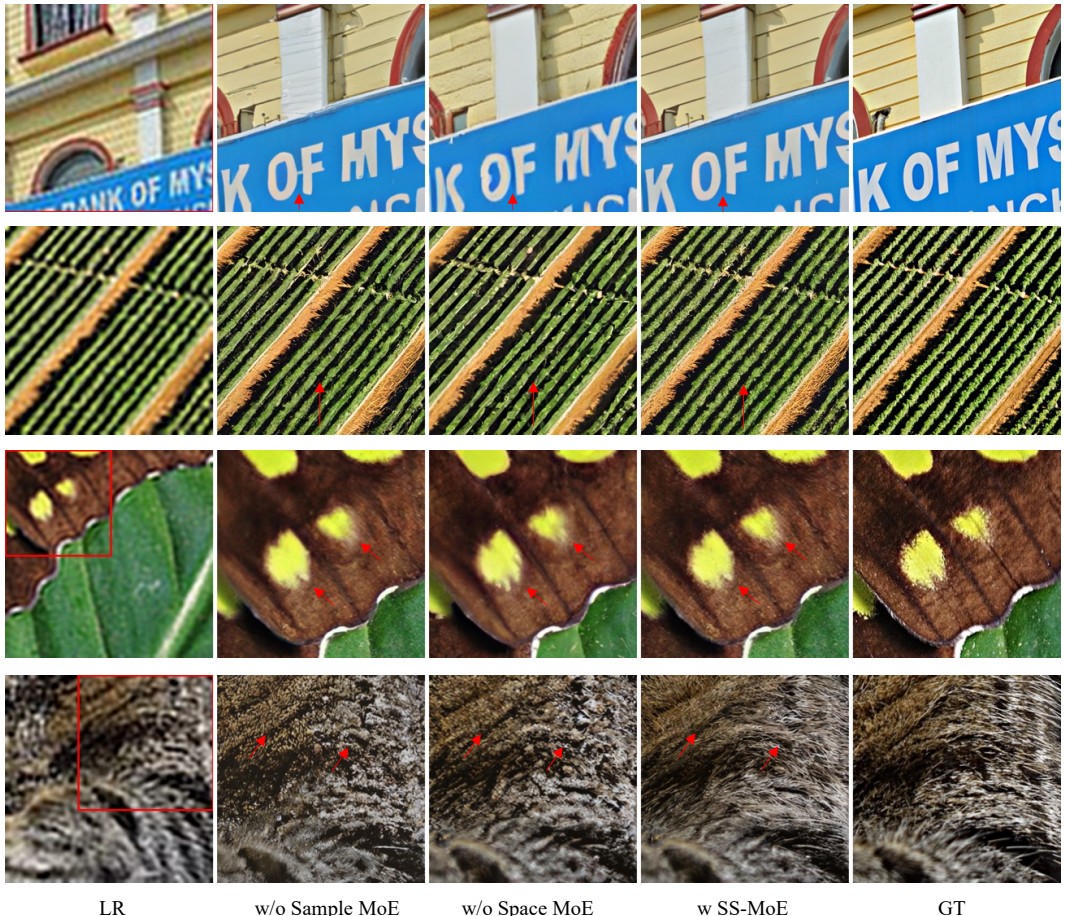

|      |             |             |          |     |
|------|-------------|-------------|----------|-----|
| LR   | w/o Sample MoE | w/o Space MoE | w SS-MoE | GT  |

Figure 8: Qualitative results of SS-MoE on $4\times$ SR. (Zoom in for the best view)

## C    QUALITATIVE RESULTS OF ABLATION STUDY

To validate the effectiveness of SS-MoE and FCD, we show qualitative results of both modules.

### C.1    QUALITATIVE RESULTS OF SS-MOE

The results are shown in Fig. 8. It can be seen that both individual Sample MoE and Space MoE generate images with distortion or over-smooth, but SS-MoE generated more realistic images with more details, which the effectiveness of both modules.

### C.2    QUALITATIVE RESULTS OF FCD

We present qualitative results of FCD in Fig. 9. It is clear that FCM can compensate images with high-frequency details, especially for human skin and animal fur. Furthermore, we conduct spectrum analysis and the results are in Fig. 10. We apply the Fourier transform to an image and then shift the spectrum to the center for better visualization. In the spectrum map, the brightness of each point indicates the energy level of the corresponding frequency component. Yellow points indicate stronger energy for the corresponding frequency, while purple points indicate weaker energy. To evaluate the spectrum gap between the generated image and ground truth, we calculate Spearman's correlation coefficients and list them under images. Better correlation coefficients show that FCM can improve images' frequency consistency through compensating details.

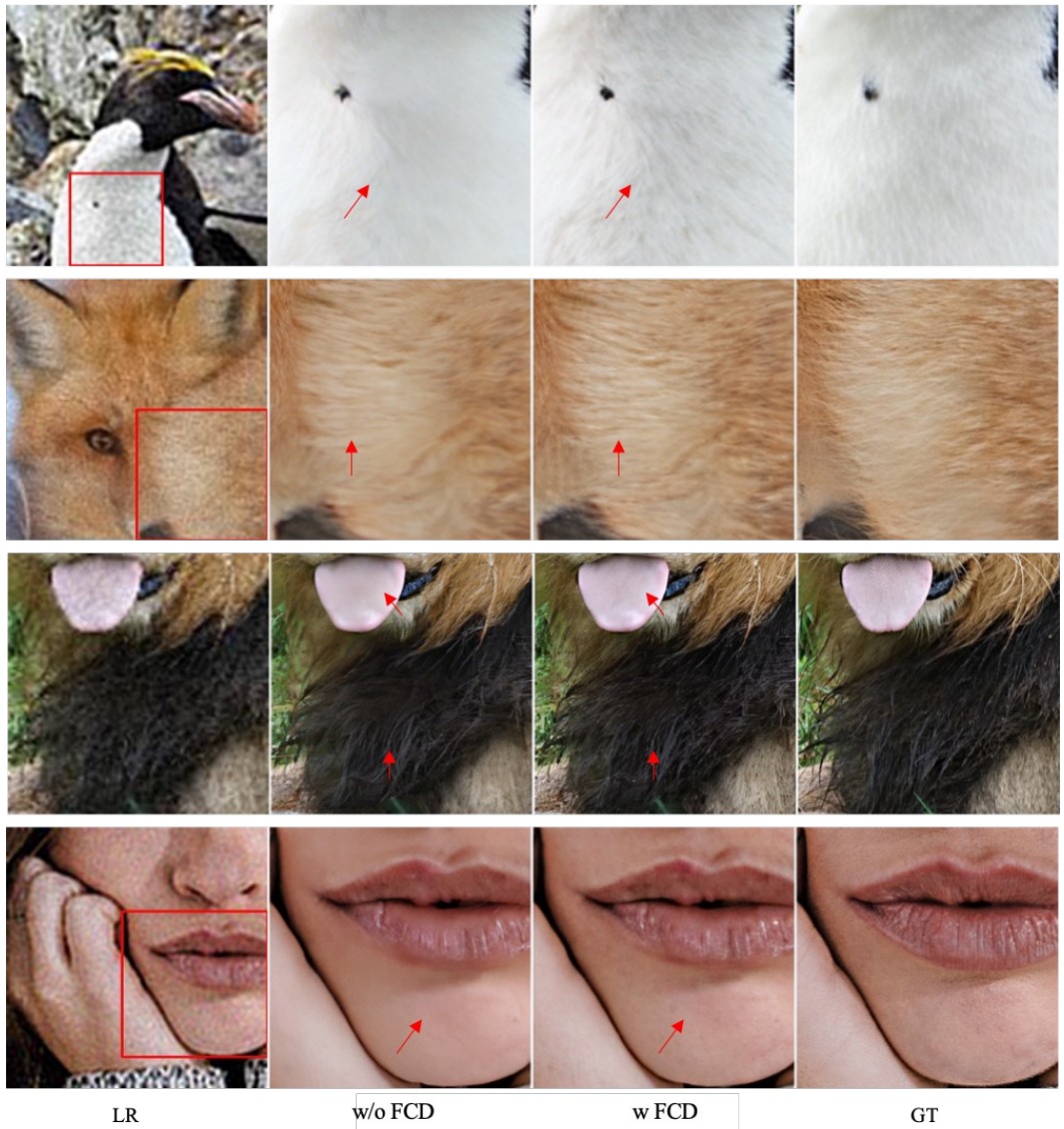

LR                  w/o FCD                  w FCD                  GT

Figure 9: Qualitative results of FCD on $4\times$ SR. (Zoom in for the best view)

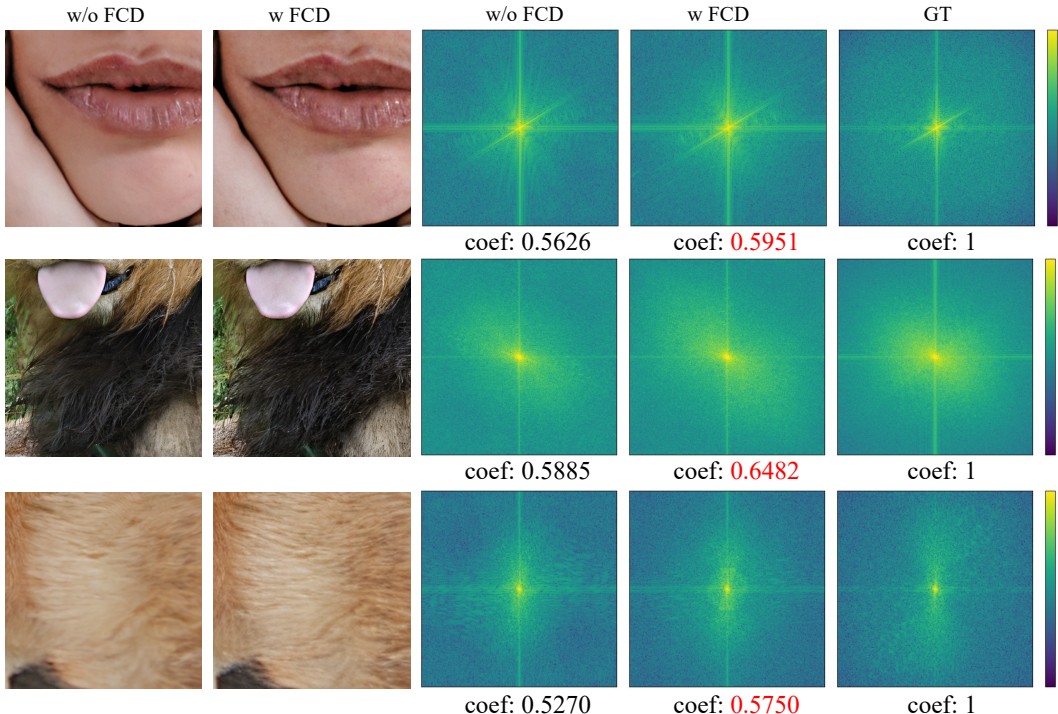

| w/o FCD | w FCD | w/o FCD | w FCD | GT |
|---------|-------|---------|-------|-----|
| | | coef: 0.5626 | coef: 0.5951 | coef: 1 |
| | | coef: 0.5885 | coef: 0.6482 | coef: 1 |
| | | coef: 0.5270 | coef: 0.5750 | coef: 1 |

Figure 10: Spectrum Analysis on FCD. (Zoom in for the best view)

Table 9: FCD on Image Reconstruction.

| Model | Coco 15 val | | Private Dataset | |
|-------|-------------|--------|-----------------|--------|
| Metrics | LPIPS↓ | FID↓ | LPIPS↓ | FID↓ |
| VAE of sd15 | 0.0746 | 17.66 | 0.0524 | 19.62 |
| +FCD | 0.0702 | 16.12 | 0.0424 | 15.71 |

## D APPLICATION OF FCD

Not only diffusion-based SR suffers from information loss caused by the compression of latent space, but image reconstruction and text-to-image generation face the same problems. We extend FCD to VAE of stable diffusion v1.5. To be specific, we add AFF-Net after the VAE decoder and use FFL Loss additionally to finetune the model. Note that lr-conditioned fuse layers are removed because there are no low-resolution images in the two tasks. The number of training steps is 7e5 and the batch size is 32. After training, we test VAE's reconstruction ability on Coco 2017 valid set (5000 images) and a private test set (1000 images collected from the Internet). Quantitative results are shown in Tab. 9. FCD shows consistent improvement on both LPIPS and FID and qualitative results in Fig. 11 align with the conclusion.

Then we apply the two VAEs to text-to-image generation. As shown in Fig. 12, it compensates image local high-frequency areas, like eyes and mouth, thus restoring the distortion compared with the original VAE. Note that, we only replace the stable diffusion's decoder with FCD, which means it is compatible with all sd v1.5 base models.

## E MORE QUALITATIVE COMPARISONS

More qualitative Comparisons among diffusion-based SR methods are as follows.

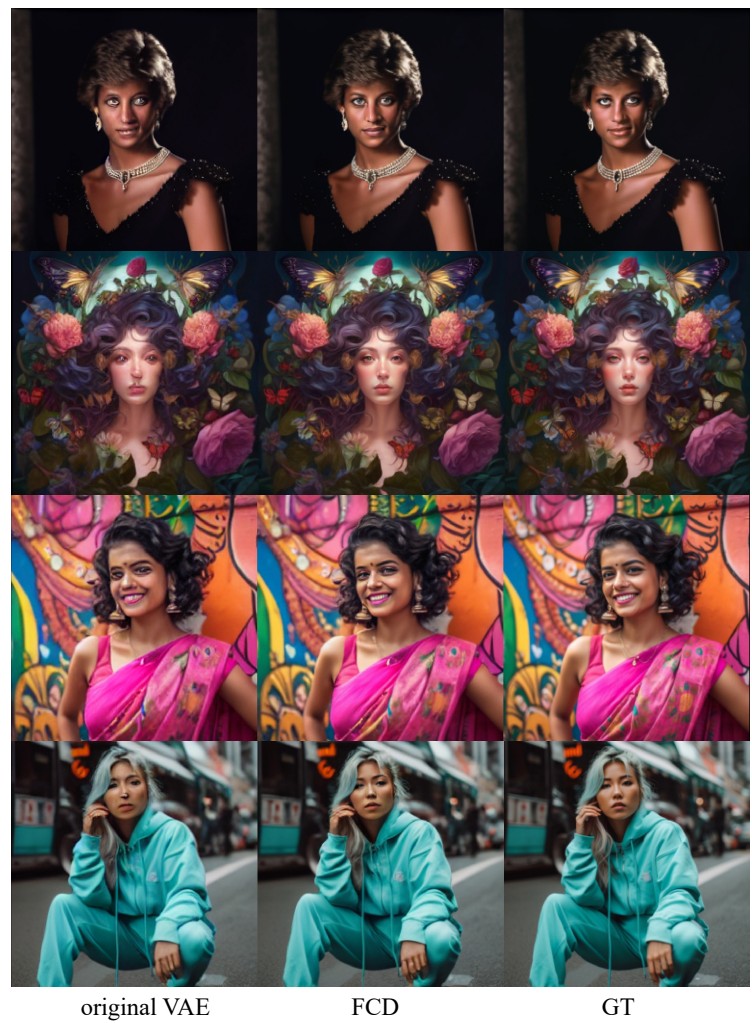

| original VAE | FCD | GT |

Figure 11: Qualitative results of FCD on image reconstruction. (Zoom in for the best view)

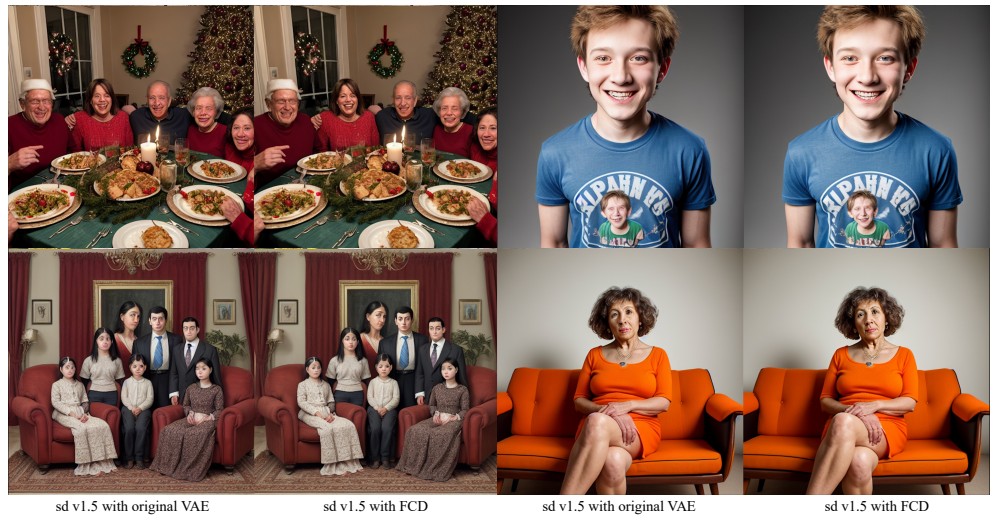

| sd v1.5 with original VAE | sd v1.5 with FCD | sd v1.5 with original VAE | sd v1.5 with FCD |

Figure 12: Qualitative results of FCD on text-to-image generation. (Zoom in for the best view)

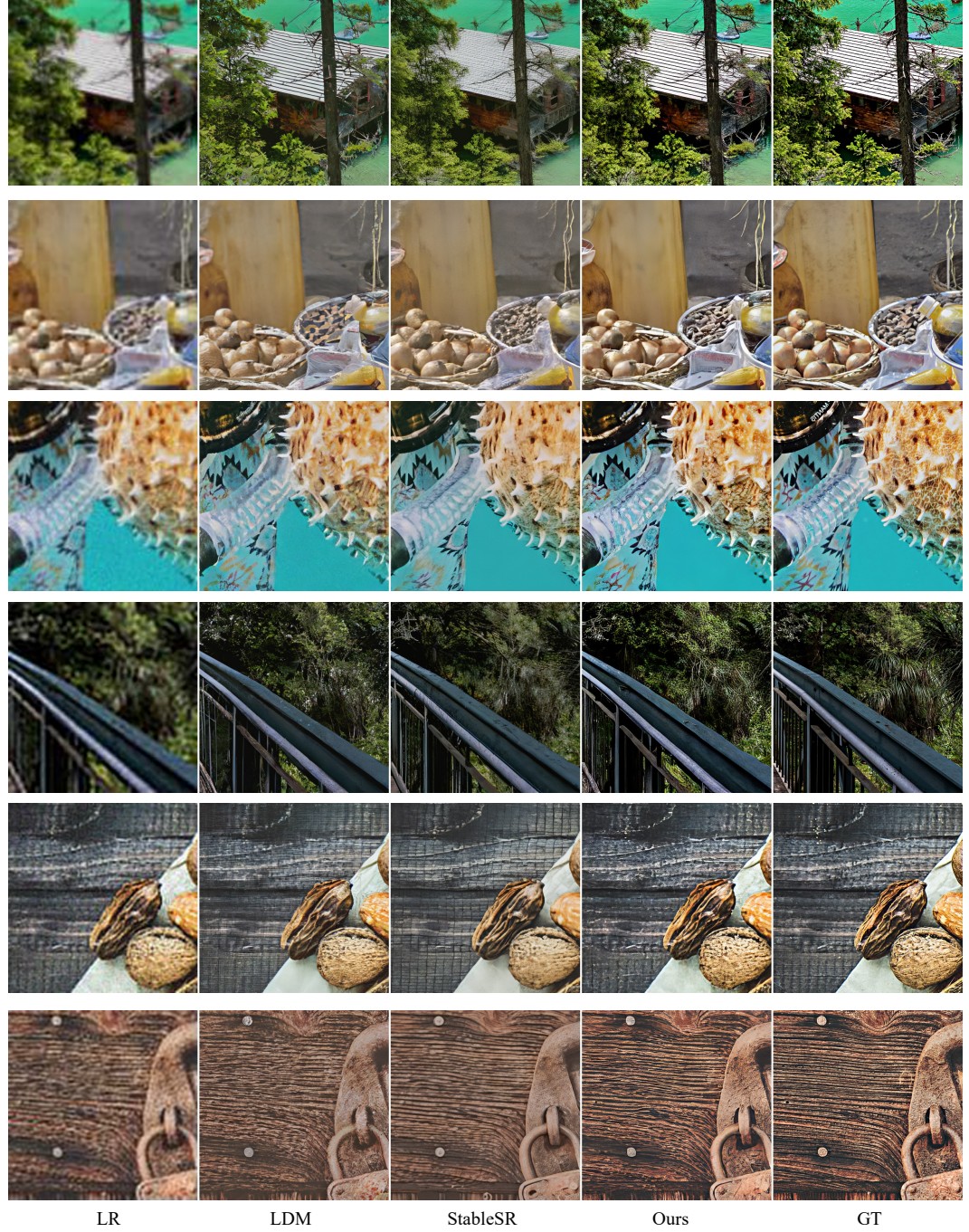

LR        LDM        StableSR        Ours        GT

Figure 13: Qualitative comparisons on $4\times$ SR ($128 \rightarrow 512$). (Zoom in for the best view)

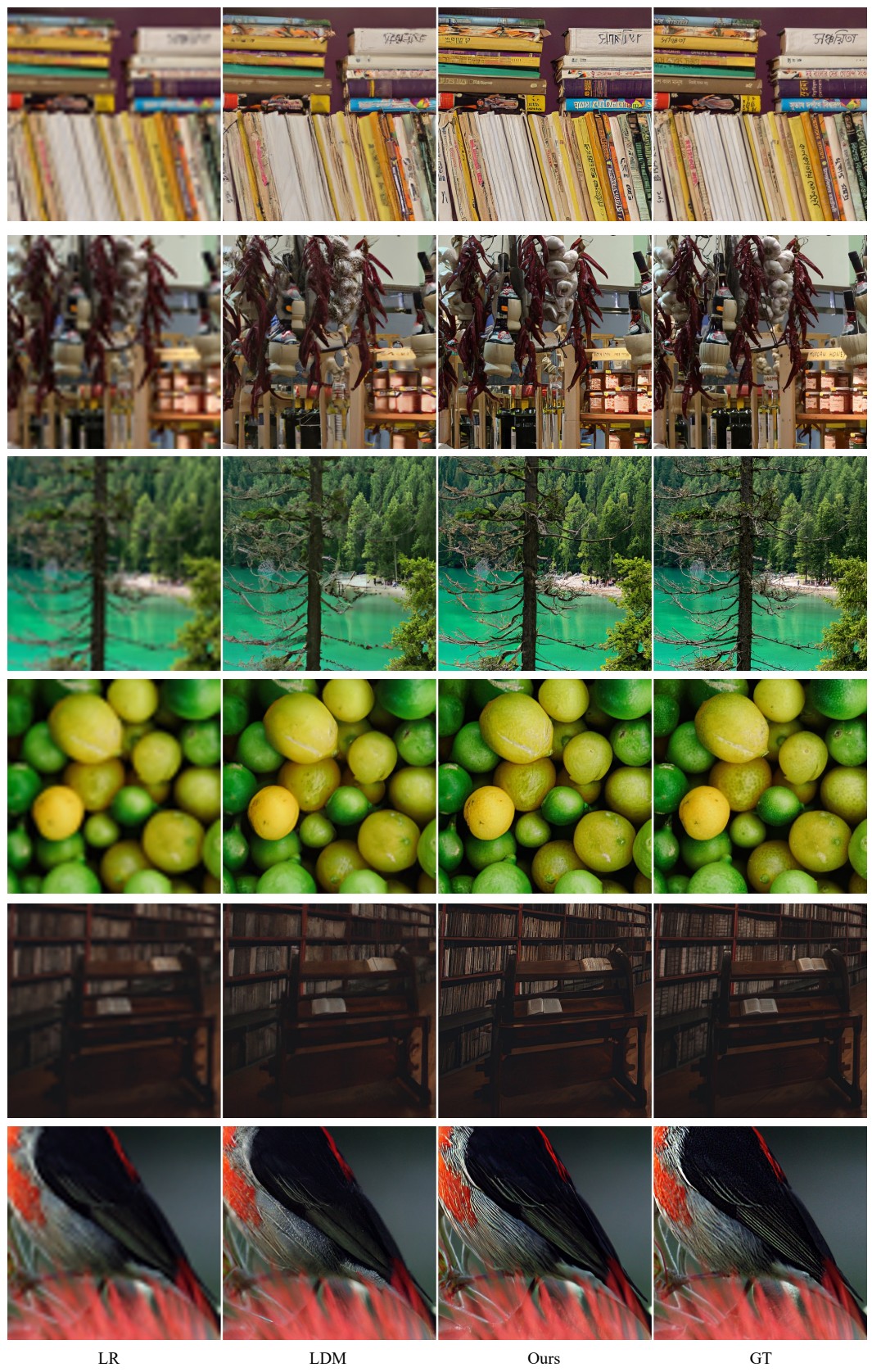

LR           LDM           Ours           GT

Figure 14: Qualitative comparisons on 8× SR (64 → 512). (Zoom in for the best view)

