# OpenReview forum: "Image Super-Resolution via Latent Diffusion: A Sampling-Space Mixture of Experts and Frequency-Augmented Decoder Approach"
_ICLR.cc/2024/Conference — Submitted to ICLR 2024_

### Official Review · Reviewer_uAFu · 2023-10-26

**Soundness:** 3 good
**Presentation:** 3 good
**Contribution:** 2 fair
**Rating:** 5
**Confidence:** 4

**Summary:**

This paper proposes a novel latent-diffusion-based framework for image super-resolution. Experiments achieve state-of-the-art of x4 and x8 super-resolution among the latest competing diffusion-based SR models.

**Strengths:**

Quantitative and qualitative results all reveal the superiority of the proposed model compared to the latest diffusion-based models.

**Weaknesses:**

1.	The point of compensating for the information loss in the autoencoder is interesting, but the novelty is limited. The newly proposed components are weak in cohesion and continuity in solving the information loss problem of the AE space and information compression.
2.	The purpose of SS-MoE seems not to be designed specifically for information loss, i.e., the motivation is not clearly written in the paper. Though the performance can be slightly upgraded with the component, the parameters are multiple times larger than the model without the component (according to Table 6 and Table 3, and the paper didn’t show the parameter increase of each component). There may be other efficient designs with the increased parameters of SS-MoE.
3.	Missing comparisons with some new baselines: DiffIR [1], DIffBIR [2], ResShift [3]

[1] Diffir: Efficient diffusion model for image restoration.
[2] DiffBIR: Towards Blind Image Restoration with Generative Diffusion Prior.
[3] Resshift: Efficient diffusion model for image super-resolution by residual shifting.

**Questions:**

Please explain in detail the motivation of SS-MoE. Others please refer to the weaknesses part.

---

> ### Author Response · Authors · 2023-11-21
> **Response to Reviewer uAFu (part 1/3)**
>
> We thank the reviewer for the constructive comments. We have incorporated your comments into the revision. We added qualitative results of ablation studies in Appendix C.1 and Appendix C.2. We would appreciate it if you have any further questions or comments.
>
> **Q1: The point of compensating for the information loss in the autoencoder is interesting, but the novelty is limited. The newly proposed components are weak in cohesion and continuity in solving the information loss problem of the AE space and information compression.**
>
> A1: Thanks for your interest in information loss in the autoencoder.
> To the best of our knowledge, **we are the first to address this problem using a trainable frequency-aware compensation network, which is simple yet effective**. Existing works[1, 2] attempt to implicitly enhance the decoder's generation ability with an additional loss, which increases the difficulty of training the decoder. In contrast, we design an additional frequency compensation network cascaded with the original decoder, which can **complement the decoder’s original ability**. To facilitate training, the convolutions of the frequency compensation network are conducted with frequency operators, making the network **frequency-aware and capable of compensating images in the frequency domain directly**. In addition to quantitative results, we have included qualitative results, spectrum analysis, and applications of the FCD to demonstrate its superiority.
>
>
>
> 1. **Quantitative results**
>
>     **We suggest focusing on LPIPS and NIQE when reviewing the qualitative results.**
>
>     **a.** To thoroughly evaluate our methods, we have included 6 metrics in the paper: (1)PSNR, SSIM(objective quality metrics) (2)LPIPS, FID, MUSIQ, NIQE( subjective quality metrics). Objective quality metrics may not always align with human perception of image quality and often contradicts subjective quality metrics[6]. **This study primarily focuses on perception-based super-resolution and prioritizes subjective metrics.**
>     **b.** As shown in Tab.5, we obtained reasonable values for all metrics. However, as **Reviewer dphS and existing works[3,4,5] have mentioned, LPIPS and NIQE are generally more convincing in real image restoration**. Therefore, we compare methods' LPIPS and NIQE to validate their effectiveness in the following part.
>     Compared with the baseline, both AFF-Net and FFL Loss improve the metrics. As shown in the following Tab.1, when using both, there are 7.3% improvement of LPIPS and 22.1% improvement of NIQE in contrast to baseline. Comparing UNet and AFF, they use the same architecture expcet that AFF uses convolution operation in frequency domain.  Better LPIPS and almost comparable NIQE show the effectiveness of frequency operation in FCD.
>
> Table1 Ablation studies of FFL Loss and AFF-Net
> | FFL Loss  | AFF-Net | LPIPS$\downarrow$  | NIQE$\downarrow$ |
> | --------  | --------| ------ | ------ |
> |           |         | 0.3038 | 5.6093 |
> |  &#10004; |         | 0.2892 | 4.5789 |
> |    |  &#10004;      | 0.2808 | 4.4337 |
> |  &#10004; |&#10004; | 0.2815 | 4.3675 |
>
> Table2 Comparsion AFF-Net and UNet
> |  Model  | LPIPS$\downarrow$  | NIQE$\downarrow$ |
> | --------| ------ | -------- |
> |  UNet+FFL| 0.2856 | 4.3652 |
> |  FCD+FFL | 0.2815 | 4.3675 |
>
> 2. **Qualitive results and spectrum analysis**
>
>     As shown in Appendix C.2, the [qualitative results](https://drive.google.com/file/d/1dvaUvtknGEjVPnD7OiDQVRLqS0PSwr1V/view?usp=sharing) illustrate that FCD effectively compensates for high-frequency details, adding realistic features such as animal fur and skin texture, thereby avoiding oversmooth. Additionally, the spectrum analysis presented in [Fig.10](https://drive.google.com/file/d/15ECHoCPny0Xx_wHPI2LLLsyxEAhGjmK3/view?usp=sharing) demonstrates that FCD generates images with a **higher correlation coefficient** between their spectrum and the spectrum of the ground truth, indicating improved frequency consistency.
>
> 3. **Application to other tasks**
>
>     We have **expanded the application of FCD to include image reconstruction and text-to-image generation**. The corresponding details are provided in Appendix D, and the results are showcased in [Fig.11](https://drive.google.com/file/d/1TWHiYD4XEvnmCQ8VkwIZXxwFifA6mfrB/view?usp=sharing) and [Fig.12](https://drive.google.com/file/d/1kLk0RGZrYVNcPiil7B02dXgKOtevPALK/view?usp=sharing). Notably, the completion of high-frequency details in human faces enhances the realism of the generated images, demonstrating the effectiveness of FCD.

---

> > ### Author Response · Authors · 2023-11-21
> > **Reference for Q1**
> >
> > **Reference**
> >
> > [1] Jiang, Liming, et al. "Focal frequency loss for image reconstruction and synthesis." Proceedings of the IEEE/CVF International Conference on Computer Vision. 2021.
> >
> > [2] Lin, Xinmiao, et al. "Catch Missing Details: Image Reconstruction with Frequency Augmented Variational Autoencoder." Proceedings of the IEEE/CVF Conference on Computer Vision and Pattern Recognition. 2023.
> >
> > [3] Wang, Zhihao, Jian Chen, and Steven CH Hoi. "Deep learning for image super-resolution: A survey." IEEE transactions on pattern analysis and machine intelligence 43.10 (2020): 3365-3387.
> >
> > [4] Li, Xin, et al. "Diffusion Models for Image Restoration and Enhancement--A Comprehensive Survey." arXiv preprint arXiv:2308.09388 (2023).
> >
> > [5] Chen, Honggang, et al. "Real-world single image super-resolution: A brief review." Information Fusion 79 (2022): 124-145.
> >
> > [6] Blau, Yochai, and Tomer Michaeli. "The perception-distortion tradeoff." Proceedings of the IEEE conference on computer vision and pattern recognition. 2018.

---

> ### Author Response · Authors · 2023-11-21
> **Response to Reviewer uAFu (part 2/3)**
>
> **Q2: The purpose of SS-MoE seems not to be designed specifically for information loss, i.e., the motivation is not clearly written in the paper.**
>
> **Q3: Please explain in detail the motivation of SS-MoE.**
>
> A3: It is right that SS-MoE is not specifically designed for addressing information loss. The primary motivation of this paper is to develop an efficient latent-based diffusion-based SR method with minimal computational overhead. To achieve this goal, we optimize the framework from two perspectives: alleviating information loss and designing an efficient network architecture.
>
> For the architecture, the high computational cost of diffusion-based SR limits the model's scalability using conventional methods like stacking more blocks or increasing channels. However, as a sequential denoising process, we can activate a specific denoising expert at different timesteps, which **incurs no additional inference cost but scales up the model**. Specifically, we propose SS-MoE, which consists of Sample MoE and Space MoE.
>
> **First, Sample MoEs can concentrate on fewer denoising steps thus learning better.** As the noisy level of timesteps changes monotonically, the same denoise expert can be used in adjacent timesteps. Therefore, we uniformly divide all timesteps into N stages and assign a single Sampling Expert to each stage. By utilizing additional experts, the number of timesteps in each block is reduced to $1000/N$, allowing each expert to enhance their focus on understanding the unique characteristics of the specific denoising steps assigned to them.
>
> **Second, Space MoEs can scale each UNet in a reparameter way.** While training we assign tokens in a random way to multiple experts with the same initialization and average their weights to one FFN during inference. But why average? **To achieve a robust and stronger network.** Without any interaction, multiple experts can be seen as subnetworks with the same initialization, each processing different data, which results in independent optimization trajectories. However, the weight sharing mechanism, i.e., equation (5) in the paper, ensures communication between experts, so their optimization trajectories are similar (but not identical). Moreover, the same initialization guarantees that all networks lie within the same basin of the error landscape[1]. It’s common for network solutions to oscillate around local optima[2], so combining the model weights can achieve a more robust network and improve accuracy.
>
> We conducted experiments to validate its effectiveness on 4x Super Resolution (SR). The results are shown in the following table. Comparing the last two rows, all metrics improved and more importantly, **the averaging operation is cost-free** to implement because it does not harm the model and need no additional computation cost during inference. Comparing the first and the third row, we can see that Space MoE indeed improves the model’s performance in terms of all metrics by enlarging the model’s scale. More quality results of SS-MoE can be seen in [Appendix C.1](https://drive.google.com/file/d/1H6VX1IvApOhvYul1QYR9mmCMpRW7GRox/view?usp=sharing).
>
> |  Model    |  Average     |  PSNR$\uparrow$     | SSIM$\uparrow$      | LPIPS$\downarrow$     | FID$\downarrow$      | MUSIQ$\uparrow$     |NIQE$\downarrow$    |
> | --------| -------- | -------- | -------- | -------- | -------- | -------- |-------- |
> |  Baseline | -  |  21.86     | 0.5554     | 0.3264     | 27.36     | 62.89 | 5.64 |
> | + Space Moe | &#10008; |  22.09     | 0.5650     | 0.3212    | 26.27     | 63.06 | 5.47 |
> | + Space Moe | &#10004;  |  22.09     | 0.5649     | 0.3201     | 26.21     | 63.14 | 5.39 |
>
>
> **Reference**
>
> [1] Wortsman, Mitchell, et al. "Model soups: averaging weights of multiple fine-tuned models improves accuracy without increasing inference time." International Conference on Machine Learning. PMLR, 2022.
>
> [2] Von Oswald, Johannes, et al. "Neural networks with late-phase weights." International Conference on Learning Representations. 2021.

---

> ### Author Response · Authors · 2023-11-21
> **Response to Reviewer uAFu (part 3/3)**
>
> **Q4: Though the performance can be slightly upgraded with the component, the parameters are multiple times larger than the model without the component (according to Table 6 and Table 3, and the paper didn’t show the parameter increase of each component). There may be other efficient designs with the increased parameters of SS-MoE.**
> A4: We have updated Tab. 6 to include the performance of the vanilla baseline and Tab. 3 to include the parameters of each component. We would like to clarify that **only one denoise expert is activated at each timestep**, so the inference cost is increased by only 3% caused by SS-MoE, even though its parameters is multiple times larger than the baseline. As shown in Tab.3, with SS-MoE, the performance on all metrics for both 4x and 8x SR has improved. So, we believe the additional computation cost is negligible, especially when compared to the 19.3% and 19.1% improvement in LPIPS score on 4x and 8x SR, respectively.
>
> As for other efficient designs with the increased parameters of SS-MoE, we think you are referring to larger models with the same parameter scale. This will definitely lead to a much higher computational cost. However, **the key advantage of SS-MoE is its ability to scale the model's parameters and capacity with negligible computational cost**.
>
>
> **Q5: Missing comparisons with some new baselines: DiffIR, DIffBIR, ResShift**
>
> A5: Thanks for your issue. According to the [ICLR 2024 Reviewer Guide](https://iclr.cc/Conferences/2024/ReviewerGuide), **the three papers are contemporaneous**(Diffir:ICCV2023 hold on October 4; DiffBIR: submited Arxiv on Aug 16; ResShift: NIPS2023 will hold in December). This is how it is explained in the guide, "We consider papers contemporaneous if they are published (available in online proceedings) within the last four months. That means, since our full paper deadline is September 28, **if a paper was published (i.e., at a peer-reviewed venue) on or after May 28, 2023, authors are not required to compare their own work to that paper.** Authors are encouraged to cite and discuss all relevant papers, but they may be excused for not knowing about papers not published in peer-reviewed conference proceedings or journals, which includes papers exclusively available on arXiv. Reviewers are encouraged to use their own good judgement and, if in doubt, discuss with their area chair. "
>
> While it's not strictly necessary to compare our method with others, we've made an effort to provide as many comparisons as possible. We've compared our method with DIffIR[1], DIffBIR[2], and ResShift[3] on the DIV2K dataset. The results are as follows. For both LPIPS and NIQE metrics, a lower value indicates better high-resolution images. Our method outperforms ResShift, achieves better NIQE than DiffIR, and better LPIPS than DiffBIR. It's important to note, however, that both DiffIR and DiffBIR are **two-stage models**, consisting of a diffusion model and either SwinIR or a pretrained encoder-decoder. Therefore, the comparison is **not entirely fair**.
>
> |  Method  | LPIPS$\downarrow$  | NIQE$\downarrow$ |
> | -------- | ------ | ---- |
> |  ResShift| 0.3452 | 6.69 |
> |  DiffIR  | 0.2592 | 5.29 |
> |  DiffBIR | 0.3692 | 4.56 |
> |  Ours    | 0.2821 | 4.72 |
>
> **Reference**
>
> [1] Xia, Bin, et al. "Diffir: Efficient diffusion model for image restoration."  Proceedings of the IEEE/CVF conference on computer vision and pattern recognition. 2023.
>
> [2] Lin, Xinqi, et al. "DiffBIR: Towards Blind Image Restoration with Generative Diffusion Prior." arXiv preprint arXiv:2308.15070 (2023).
>
> [3] Yue, Zongsheng, Jianyi Wang, and Chen Change Loy. "Resshift: Efficient diffusion model for image super-resolution by residual shifting." 37th Conference on Neural Information Processing Systems. 2024.

---

### Official Review · Reviewer_krWG · 2023-10-30

**Soundness:** 2 fair
**Presentation:** 3 good
**Contribution:** 2 fair
**Rating:** 3
**Confidence:** 5

**Summary:**

This paper summarizes two issues in using the diffusion model to address image super-resolution, namely distortion caused by the compression of latent space and the huge computational cost. This work proposes a frequency compensation module to enhance
the frequency components and use Sample-Space Mixture of Experts (SS-MoE) to improve the capacity of the SR model. The visual results provided in the paper appear to have clearer details compared to other methods, and it also demonstrates certain advantages in some quantitative metrics.

**Strengths:**

1. The paper provides a comprehensive and clear analysis and summary of the current methods and issues in super-resolution.

**Weaknesses:**

1. The results in the paper do not stand out significantly when compared to other methods, and they are not the best in terms of the metrics, making it challenging to demonstrate the superiority of this method over other super-resolution methods.
2. The reason for designing SS-MoE in this way is not explicitly explained. Initially, multiple MoEs were designed to separately handle different noise sources. However, during inference, an averaging step is performed to parameterize the weights, which conflicts with the original motivation.
3. In the experimental ablation study of FCD, the results show variations in the metrics, with some being high and others low. This inconsistency in the results makes it difficult to determine whether the approach used in the paper is the best one.
4. There are several typographical errors in this paper. I recommend conducting a more thorough proofreading.

**Questions:**

Refer to weakness

---

> ### Author Response · Authors · 2023-11-21
> **Response to Reviewer krWG (part 1/3)**
>
> We thank the reviewer for the constructive comments. We have incorporated your comments into the revision and have modified typographical errors in this paper. We would appreciate it if you have any further questions or comments.
>
>
> **Q1: The results in the paper do not stand out significantly when compared to other methods, and they are not the best in terms of the metrics, making it challenging to demonstrate the superiority of this method over other super-resolution methods.**
> A1: Thanks for your question. We want to discuss the complexity of metrics and then we make an analysis of our metric improvement in detail and compare it with accepted papers that are about the same topic(include a paper in ICLR 2023). We do believe the results can show the superiority of our method undoubtedly.
>
> 1. **We suggest focusing on LPIPS and NIQE when reviewing the qualitative results.**
>
>     **a.** To thoroughly evaluate our methods, we have included 6 metrics in the paper: PSNR, SSIM, LPIPS, FID, MUSIQ, and NIQE. Among them, objective quality metrics(PSNR and SSIM) may not always align with human perception of image quality and often contradicts subjective quality metrics(LPIPS, FID, MUSIQ, and NIQE)[11]. **This study primarily focuses on perception-based super-resolution and prioritizes subjective metrics.**
>
>     **b.** However, Designing metrics for photo-realistic SR that align with human visual perception is challenging and is still an ongoing research area[1,2]. Consequently, different papers may report different metrics, and **no single method has been found to excel in all subjective quality metrics as shown in the [figure](https://drive.google.com/file/d/1Nl6p6LGRhYCukMtrrb4n7T0P3sgMq8Mq/view?usp=sharing)**[3].
>
>     **c.** LPIPS and FID are reference-based metrics that assess perceptual quality. LPIPS specifically evaluates perceptual consistency at the image level, while FID measures the overall distributional differences between real and generated images. Therefore, FID captures the overall style difference between two image folders rather than image-level similarity. On the other hand, MUSIQ and NIQE are non-reference metrics that evaluate image quality. MUSIQ evaluates image quality with multi-scale input, while NIQE assesses the naturalness of the image by analyzing statistical features. As shown in Tab.1, we obtained reasonable values for all metrics. However, as **Reviewer dphS and existing works[4,5,6] have mentioned, LPIPS and NIQE are generally more convincing in real image restoration**. Therefore, we compare methods' LPIPS and NIQE to validate their effectiveness in the following part.
>
>
> 2. **Quantitative results.** In Tab.1 of the paper, we achieved state-of-the-art results in terms of LPIPS and NIQE on the DIV2K dataset. Additionally, our method outperformed existing diffusion-based approaches on RealSR and DRealSR. The concludions align with the result of user study in [Appendix B](https://drive.google.com/file/d/17IKksV6i6Szk8NaqCSgEE6TTzOtrOlxh/view?usp=sharing).
>
>
> 3. **We believe the performance improvement in Table 1 of the paper is not trivial.** There are excellent accepted papers[7,8,9,10] with similar and even less metric improvement. In KDSR[7](ICLR 2023), there are 0.0189 and 0.0248 LPIPS improvement on two real-world SR benchmarks. In AMNet-RL[9](CVPR 2021), there are 0.23, -0.12, 0.06 and 0.06 NIQE improvement on four SR benchmarks. We list metrics of diffusion-based methods in the following table and present our metric improvement with the existing best metric in the last row. Compared with the works we mentioned before, our metric improvement is stronger.
>
> | Method | DIV2K |      | RealSR|      | DRealSR |     |
> | -------| ------| -----|  -----|------| -----|  -----|
> | Metric       | LPIPS$\downarrow$| NIQE$\downarrow$|  LPIPS$\downarrow$| NIQE$\downarrow$| LPIPS$\downarrow$| NIQE$\downarrow$|
> | LDM      | 0.3264 | 5.64 | 0.3159 | 6.78 | 0.3379 | 7.37 |
> | StableSR | 0.3143 | 4.81 | 0.2974 | 6.35 | 0.3353 | 6.89 |
> | DIV2K    | 0.2821 | 4.72 | 0.2719 | 5.96 | 0.3017 | 6.89 |
> |improvement|0.0322 | 0.09 | 0.0255 | 0.39 | 0.0336 | 0.00 |

---

> > ### Author Response · Authors · 2023-11-21
> > **Reference for Q1**
> >
> > **Reference**
> >
> > [1] Jiang, Qiuping, et al. "Single image super-resolution quality assessment: a real-world dataset, subjective studies, and an objective metric." IEEE Transactions on Image Processing 31 (2022): 2279-2294.
> >
> > [2] Zhang, Wenlong, et al. "SEAL: A Framework for Systematic Evaluation of Real-World Super-Resolution." arXiv preprint arXiv:2309.03020 (2023).
> >
> > [3] Li, Chongyi, et al. "Embedding fourier for ultra-high-definition low-light image enhancement." International Conference on Learning Representations. 2023.
> >
> > [4] Wang, Zhihao, Jian Chen, and Steven CH Hoi. "Deep learning for image super-resolution: A survey." IEEE transactions on pattern analysis and machine intelligence 43.10 (2020): 3365-3387.
> >
> > [5] Li, Xin, et al. "Diffusion Models for Image Restoration and Enhancement--A Comprehensive Survey." arXiv preprint arXiv:2308.09388 (2023).
> >
> > [6] Chen, Honggang, et al. "Real-world single image super-resolution: A brief review." Information Fusion 79 (2022): 124-145.
> >
> > [7] Xia, Bin, et al. "Knowledge distillation based degradation estimation for blind super-resolution." International Conference on Learning Representations. 2023.
> >
> > [8] Park, Seobin, et al. "Learning Controllable Degradation for Real-World Super-Resolution via Constrained Flows." International Conference on Machine Learning. 2023.
> >
> > [9] Hui, Zheng, et al. "Learning the non-differentiable optimization for blind super-resolution." Proceedings of the IEEE/CVF conference on computer vision and pattern recognition. 2021.
> >
> > [10] Zhang, Wenlong, et al. "Ranksrgan: Super resolution generative adversarial networks with learning to rank." IEEE Transactions on Pattern Analysis and Machine Intelligence 44.10 (2021): 7149-7166.
> >
> > [11] Blau, Yochai, and Tomer Michaeli. "The perception-distortion tradeoff." Proceedings of the IEEE conference on computer vision and pattern recognition. 2018.

---

> ### Author Response · Authors · 2023-11-21
> **Response to Reviewer krWG (part 2/3)**
>
> **Q2: The reason for designing SS-MoE in this way is not explicitly explained. Initially, multiple MoEs were designed to separately handle different noise sources. However, during inference, an averaging step is performed to parameterize the weights, which conflicts with the original motivation.**
> A2: The overall motivation of SS-MoE is to **scale the SR model to achieve performance improvement without increasing inference overhead given the huge cost of diffusion-based SR**. As we discussed in Sec.3.2, SS-MoE consists of Sample MoE and Space MoE.
>
> 1. Multiple Sample MoEs are designed to denoise for images with varying levels of noise intensity in the whole reverse diffusion process. So there are several denoise UNet with the same structure trained and used at different denoise stages.
> 2. We design Space MoE in each denoise UNet's FFN to scale the capacity of networks efficiently. To be specific, while training we randomly assign tokens to multiple experts with the same initialization instead of assigning by routing network which will cause additional computation costs. But Why average?
>
>     **a. To achieve a robust and stronger network.** Without any interaction, multiple experts can be seen as subnetworks with the same initialization, each processing different data, which results in independent optimization trajectories. However, the weight sharing mechanism, i.e., equation (5) in the paper, ensures communication between experts, so their optimization trajectories are similar (but not identical). Moreover, the same initialization guarantees that all networks lie within the same basin of the error landscape[1]. It's common for network solutions to oscillate around local optima[2], so combining the model weights can achieve a more robust network and improve accuracy.
>
>     **b.** We conducted experiments to validate its effectiveness on 4x Super Resolution (SR). The results are shown in the following table. Comparing the last two rows, all metrics improved and more importantly, **the averaging operation is cost-free** to implement because it does not harm the model and need no additional computation cost during inference. Comparing the first and the third row, we can see that Space MoE indeed improves the model’s performance in terms of all metrics by enlarging the model’s scale. More quality results of SS-MoE can be seen in [Appendix C.1](https://drive.google.com/file/d/1H6VX1IvApOhvYul1QYR9mmCMpRW7GRox/view?usp=sharing).
>
> |  Model    |  Average     |  PSNR$\uparrow$     | SSIM$\uparrow$      | LPIPS$\downarrow$     | FID$\downarrow$      | MUSIQ$\uparrow$     |NIQE$\downarrow$    |
> | --------| -------- | -------- | -------- | -------- | -------- | -------- |-------- |
> |  Baseline | -  |  21.86     | 0.5554     | 0.3264     | 27.36     | 62.89 | 5.64 |
> | + Space Moe | &#10008; |  22.09     | 0.5650     | 0.3212    | 26.27     | 63.06 | 5.47 |
> | + Space Moe | &#10004;  |  22.09     | 0.5649     | 0.3201     | 26.21     | 63.14 | 5.39 |
>
> **Reference**
>
>
> [1] Wortsman, Mitchell, et al. "Model soups: averaging weights of multiple fine-tuned models improves accuracy without increasing inference time." International Conference on Machine Learning. PMLR, 2022.
>
>
> [2] Von Oswald, Johannes, et al. "Neural networks with late-phase weights." International Conference on Learning Representations. 2021.

---

> ### Author Response · Authors · 2023-11-21
> **Response to Reviewer krWG (part 3/3)**
>
> **Q3: In the experimental ablation study of FCD, the results show variations in the metrics, with some being high and others low. This inconsistency in the results makes it difficult to determine whether the approach used in the paper is the best one.**
>
> A3: We will demonstrate the effectiveness of FCD via quantitative results, qualitative results, and the application of FCD to other tasks.
>
> We have achieved reasonable metrics in Tab.5. However, as we have stated in Q1, we will focus on LPIPS and NIQE for comparison.
> **Quantitative comparison.** In the following Tab.1, both AFF-Net and FFL Loss show improvements in these metrics compared to the baseline. When both are used together, there is a 7.3% improvement in LPIPS and a 22.1% improvement in NIQE compared to the baseline. In the following Tab.2, comparing UNet and AFF, they have the same architecture except that AFF utilizes convolution operations in the frequency domain. The better LPIPS and nearly comparable NIQE scores indicate the effectiveness of frequency operations in FCD.
>
> Table1 Ablation studies of FFL Loss and AFF-Net
> | FFL Loss  | AFF-Net | LPIPS$\downarrow$  | NIQE$\downarrow$ |
> | --------  | --------| ------ | ------ |
> |           |         | 0.3038 | 5.6093 |
> |  &#10004; |         | 0.2892 | 4.5789 |
> |    |  &#10004;      | 0.2808 | 4.4337 |
> |  &#10004; |&#10004; | 0.2815 | 4.3675 |
>
> Table2 Comparsion AFF-Net and UNet
> |  Model  | LPIPS$\downarrow$  | NIQE$\downarrow$ |
> | --------| ------ | -------- |
> |  UNet+FFL| 0.2856 | 4.3652 |
> |  FCD+FFL | 0.2815 | 4.3675 |
>
> **Qualitative results and spectrum analysis.** We also add qualitative results and spectrum analysis in Appendix C.2. The [qualitative results](https://drive.google.com/file/d/1dvaUvtknGEjVPnD7OiDQVRLqS0PSwr1V/view?usp=sharing) illustrate that FCD effectively compensates for high-frequency details, adding realistic features such as animal fur and skin texture, thereby avoiding oversmooth. Additionally, the spectrum analysis presented in [Fig.10](https://drive.google.com/file/d/15ECHoCPny0Xx_wHPI2LLLsyxEAhGjmK3/view?usp=sharing) demonstrates that FCD generates images with a **higher correlation coefficient** between their spectrum and the spectrum of the ground truth, indicating improved frequency consistency.
>
> **Application to other tasks.** We have expanded the application of FCD to include image reconstruction and text-to-image generation. The corresponding details are provided in Appendix D, and the results are showcased in [Fig.11](https://drive.google.com/file/d/1TWHiYD4XEvnmCQ8VkwIZXxwFifA6mfrB/view?usp=sharing) and [Fig.12](https://drive.google.com/file/d/1kLk0RGZrYVNcPiil7B02dXgKOtevPALK/view?usp=sharing). Notably, the completion of high-frequency details in human faces enhances the realism of the generated images, effectively demonstrating the effectiveness of FCD.
>
>
> **Q4: There are several typographical errors in this paper. I recommend conducting a more thorough proofreading.**
> A4: Thanks for your reminder. We have modified typographical errors in this paper.

---

### Official Review · Reviewer_MvoB · 2023-10-30

**Soundness:** 2 fair
**Presentation:** 2 fair
**Contribution:** 2 fair
**Rating:** 3
**Confidence:** 4

**Summary:**

In this paper, the authors proposed a new diffusion-based SR approach, the authors propose a sample-space mixture of experts strategy to improve the sampling quality and propose a frequency compensation module to reduce high-frequency reonstruction distortion.

**Strengths:**

1) The author introduce sampling-space MOE to improve the image quality of diffusion-based SR, and provide detailed ablation experiments to validate the effectiveness of the adopted strategy.
2) The authors propose a frequency loss to emphasize high-frequency distortion.
3) The authors validated the proposed method on several benchmark datasets.

**Weaknesses:**

1) The novelty of this paper is not significant. The major framework and the major modifications were proposed in other works and the authors just combine these method to establish a new method. Using the MOE strategy to enhance sampling quality and introduce high-frequency loss to enhance SR results are straight-forward operations and the authors just combine the two methods without any modification.
2) To evaluate photo-realistic SR, it is highly suggested to conduct subjective evaluation to compare different methods.
3) Based on Table 1, the advantage of the proposed method over the competing approaches are not significant.
4) The authors utilized 5 or 6 metrics to evaluate different methods, but did not discuss the results carefully. The numbers in Table 5 can not clearly validate the effectiveness of the proposed FFL and AFF-Net.

**Questions:**

Please refer to the weakness part.

---

> ### Author Response · Authors · 2023-11-21
> **Response to Reviewer MvoB (part 1/3)**
>
> We thank the reviewer for the constructive comments. We have incorporated your comments into the revision. We added user study in Appendix B. We would appreciate it if you have any further questions or comments.
>
> **Q1: The novelty of this paper is not significant.**
> A1:
> Thank you for raising the concern. We would like to clarify the novelty of our approach, which involves an explicit frequency compensation decoder and a Sample-Space MoE that efficiently scales the model throughout global sampling and individual steps.
>
> 1. **Our approach explicitly compensates for high-frequency information using a trainable, frequency-aware compensation network, rather than merely a loss function.**
>     **a.** Existing works[1,2] implicitly enhance the decoder's generative ability with an additional loss, which complicates the training of the decoder. In contrast, our proposed frequency compensation network, when cascaded with the original decoder, **complements the decoder's capabilities**. To facilitate training, the convolutions of the frequency compensation network are conducted with frequency operators, making the network frequency-aware. This allows it to **directly compensate images in the frequency domain**, simplifying the training process.
>
>     **b.** Our ablation study of the FCD demonstrates that the explicit compensation network, i.e., AFF, improves performance across all perception-based metrics compared to a model trained solely with FFL Loss. This underscores the superiority of explicit compensation. When compared to UNet, AFF achieves better LPIPS and comparable NIQE, highlighting the importance of frequency operations.
>
>     **c.** We also demonstrate the **versatility of our frequency-aware compensation network** by extending it to **text-to-image generation and image reconstruction**. As shown in [Fig.11](https://drive.google.com/file/d/1TWHiYD4XEvnmCQ8VkwIZXxwFifA6mfrB/view?usp=sharing) and [Fig.12](https://drive.google.com/file/d/1kLk0RGZrYVNcPiil7B02dXgKOtevPALK/view?usp=sharing) of Appendix D, it effectively mitigates facial distortions caused by the loss of high-frequency information in stable diffusion and VAE, showcasing its robust compensatory capabilities.
>
>
> 2. **We explore MoE for each timestep in an efficient way, not only for different sampling stages.**
>     **a.** Unlike existing MoE methods for the diffusion model[3,4], which design multiple experts for different time steps, we also explore how to scale the model within a single step. Specifically, we train several Feed-Forward Networks(FFNs) without any gate network by randomly splitting tokens. During inference, we average the weights of FFN layers at the same position to generate a robust and more powerful FFN. This approach **requires no additional inference cost**. Furthermore, while there are existing works[5,6] that use MoE for super-resolution, these typically attempt to handle different types of degradation with different experts, leading to increased computational costs.
>
>     **b.** But why average? **To achieve a robust and stronger network.** Without any interaction, multiple experts can be seen as subnetworks with the same initialization, each processing different data, which results in independent optimization trajectories. However, the weight sharing mechanism, i.e., equation (5) in the paper, ensures communication between experts, so their optimization trajectories are similar (but not identical). Moreover, the same initialization guarantees that all networks lie within the same basin of the error landscape[7]. It's common for network solutions to oscillate around local optima[8], so combining the model weights can achieve a more robust network and improve accuracy.
>
>     **c.** We conducted experiments to validate its effectiveness on 4x Super Resolution (SR). The results are shown in the following table. Comparing the last two rows, all metrics improved and more importantly, **the averaging operation is cost-free** to implement because it does not harm the model and need no additional computation cost during inference. Comparing the first and the third row, we can see that Space MoE indeed improves the model’s performance in terms of all metrics by enlarging the model’s scale. More quality results of SS-MoE can be seen in [Appendix C.1](https://drive.google.com/file/d/1H6VX1IvApOhvYul1QYR9mmCMpRW7GRox/view?usp=sharing).
>
> |  Model    |  Average     |  PSNR$\uparrow$     | SSIM$\uparrow$      | LPIPS$\downarrow$     | FID$\downarrow$      | MUSIQ$\uparrow$     |NIQE$\downarrow$    |
> | --------| -------- | -------- | -------- | -------- | -------- | -------- |-------- |
> |  Baseline | -  |  21.86     | 0.5554     | 0.3264     | 27.36     | 62.89 | 5.64 |
> | + Space Moe | &#10008; |  22.09     | 0.5650     | 0.3212    | 26.27     | 63.06 | 5.47 |
> | + Space Moe | &#10004;  |  22.09     | 0.5649     | 0.3201     | 26.21     | 63.14 | 5.39 |

---

> > ### Author Response · Authors · 2023-11-21
> > **Reference for Q1**
> >
> > **Reference**
> >
> >
> > [1] Jiang, Liming, et al. "Focal frequency loss for image reconstruction and synthesis." Proceedings of the IEEE/CVF International Conference on Computer Vision. 2021.
> >
> >
> > [2] Lin, Xinmiao, et al. "Catch Missing Details: Image Reconstruction with Frequency Augmented Variational Autoencoder." Proceedings of the IEEE/CVF Conference on Computer Vision and Pattern Recognition. 2023.
> >
> >
> > [3] Feng, Zhida, et al. "ERNIE-ViLG 2.0: Improving text-to-image diffusion model with knowledge-enhanced mixture-of-denoising-experts." Proceedings of the IEEE/CVF Conference on Computer Vision and Pattern Recognition. 2023.
> >
> >
> > [4] Balaji, Yogesh, et al. "ediffi: Text-to-image diffusion models with an ensemble of expert denoisers." arXiv preprint arXiv:2211.01324 (2022).
> >
> >
> > [5] Kim, Sijin, Namhyuk Ahn, and Kyung-Ah Sohn. "Restoring spatially-heterogeneous distortions using mixture of experts network." Proceedings of the Asian Conference on Computer Vision. 2020.
> >
> >
> > [6] Emad, Mohammad, Maurice Peemen, and Henk Corporaal. "MoESR: blind super-resolution using kernel-aware mixture of experts." Proceedings of the IEEE/CVF Winter Conference on Applications of Computer Vision. 2022.
> >
> >
> > [7] Wortsman, Mitchell, et al. "Model soups: averaging weights of multiple fine-tuned models improves accuracy without increasing inference time." International Conference on Machine Learning. PMLR, 2022.
> >
> >
> > [8] Von Oswald, Johannes, et al. "Neural networks with late-phase weights." International Conference on Learning Representations. 2021.

---

> ### Author Response · Authors · 2023-11-21
> **Response to Reviewer MvoB (part 2/3)**
>
> **Q2: To evaluate photo-realistic SR, it is highly suggested to conduct subjective evaluation to compare different methods.**
> A2: Thank you for your suggestion. Despite the time constraint, we managed to include a user study in Appendix B. We conducted this study on 210 real-world Low Resolution (LR) images from DIV2K, RealSR, and DRealSR datasets, with 17 users participating. The comparison was conducted in pairs; given an LR image as a reference, the participant was asked to choose the better High Resolution (HR) image generated either by our method or others. As shown in [Fig.6](https://drive.google.com/file/d/17IKksV6i6Szk8NaqCSgEE6TTzOtrOlxh/view?usp=sharing), **our method significantly outperforms all 7 counterparts, consistently gaining more than 69.6% of the votes.** This indicates the substantial superiority of our method in terms of human perception.
>
>
> **Q3: Based on Table 1, the advantage of the proposed method over the competing approaches are not significant.**
> Thanks for your comment. We want to point out the complexity of SR metrcis and then make analysis of quantitative results.
> A3:
> 1. **We suggest focusing on LPIPS and NIQE when reviewing the qualitative results.**
>
>     a. To thoroughly evaluate our methods, we have included 6 metrics in the paper: PSNR, SSIM, LPIPS, FID, MUSIQ, and NIQE. Among them, objective quality metrics(PSNR and SSIM) may not always align with human perception of image quality and often contradicts subjective quality metrics(LPIPS, FID, MUSIQ, and NIQE)[11]. **This study primarily focuses on perception-based super-resolution and prioritizes subjective metrics.**
>
>     b. However, Designing metrics for photo-realistic SR that align with human visual perception is challenging and is still an ongoing research area[1,2]. Consequently, different papers may report different metrics, and **no single method has been found to excel in all subjective quality metrics as shown in the [figure](https://drive.google.com/file/d/1Nl6p6LGRhYCukMtrrb4n7T0P3sgMq8Mq/view?usp=sharing)**[3].
>
>     c. LPIPS and FID are reference-based metrics that assess perceptual quality. LPIPS specifically evaluates perceptual consistency at the image level, while FID measures the overall distributional differences between real and generated images. Therefore, FID captures the overall style difference between two image folders rather than image-level similarity. On the other hand, MUSIQ and NIQE are non-reference metrics that evaluate image quality. MUSIQ evaluates image quality with multi-scale input, while NIQE assesses the naturalness of the image by analyzing statistical features. As shown in Tab.1, we obtained reasonable values for all metrics. However, as **Reviewer dphS and existing works[4,5,6] have mentioned, LPIPS and NIQE are generally more convincing in real image restoration**. Therefore, we compare methods' LPIPS and NIQE to validate their effectiveness in the following part.
>
>
> 2. **Quantitative results.** In Tab.1 of the paper, we achieved state-of-the-art results in terms of LPIPS and NIQE on the DIV2K dataset. Additionally, our method outperformed existing diffusion-based approaches on RealSR and DRealSR. The concludions align with the result of user study in [Appendix B]((https://drive.google.com/file/d/17IKksV6i6Szk8NaqCSgEE6TTzOtrOlxh/view?usp=sharing)).
>
> 3. **We believe the performance improvement in Table 1 of the paper is not trivial.** There are excellent accepted papers[7,8,9,10] with similar and even less metric improvement. In KDSR[7](ICLR 2023), there are 0.0189 and 0.0248 LPIPS improvement on two real-world SR benchmarks. In AMNet-RL[9](CVPR 2021), there are 0.23, -0.12, 0.06 and 0.06 NIQE improvement on four SR benchmarks. We list metrics of diffusion-based methods in the following table and present our metric improvement with the existing best metric in the last row. Compared with the works we mentioned before, our metric improvement is stronger.
>
> | Method | DIV2K |      | RealSR|      | DRealSR |     |
> | -------| ------| -----|  -----|------| -----|  -----|
> | Metric       | LPIPS$\downarrow$| NIQE$\downarrow$|  LPIPS$\downarrow$| NIQE$\downarrow$| LPIPS$\downarrow$| NIQE$\downarrow$|
> | LDM      | 0.3264 | 5.64 | 0.3159 | 6.78 | 0.3379 | 7.37 |
> | StableSR | 0.3143 | 4.81 | 0.2974 | 6.35 | 0.3353 | 6.89 |
> | DIV2K    | 0.2821 | 4.72 | 0.2719 | 5.96 | 0.3017 | 6.89 |
> |improvement|0.0322 | 0.09 | 0.0255 | 0.39 | 0.0336 | 0.00 |

---

> > ### Author Response · Authors · 2023-11-21
> > **Reference for Q3**
> >
> > **Reference**
> >
> >
> > [1] Jiang, Qiuping, et al. "Single image super-resolution quality assessment: a real-world dataset, subjective studies, and an objective metric." IEEE Transactions on Image Processing 31 (2022): 2279-2294.
> >
> >
> > [2] Zhang, Wenlong, et al. "SEAL: A Framework for Systematic Evaluation of Real-World Super-Resolution." arXiv preprint arXiv:2309.03020 (2023).
> >
> >
> > [3] Li, Chongyi, et al. "Embedding fourier for ultra-high-definition low-light image enhancement." International Conference on Learning Representations. 2023.
> >
> >
> > [4] Wang, Zhihao, Jian Chen, and Steven CH Hoi. "Deep learning for image super-resolution: A survey." IEEE transactions on pattern analysis and machine intelligence 43.10 (2020): 3365-3387.
> >
> >
> > [5] Li, Xin, et al. "Diffusion Models for Image Restoration and Enhancement--A Comprehensive Survey." arXiv preprint arXiv:2308.09388 (2023).
> >
> >
> > [6] Chen, Honggang, et al. "Real-world single image super-resolution: A brief review." Information Fusion 79 (2022): 124-145.
> >
> > [7] Xia, Bin, et al. "Knowledge distillation based degradation estimation for blind super-resolution." International Conference on Learning Representations. 2023.
> > [8] Park, Seobin, et al. "Learning Controllable Degradation for Real-World Super-Resolution via Constrained Flows." International Conference on Machine Learning. 2023.
> >
> > [9] Hui, Zheng, et al. "Learning the non-differentiable optimization for blind super-resolution." Proceedings of the IEEE/CVF conference on computer vision and pattern recognition. 2021.
> >
> > [10] Zhang, Wenlong, et al. "Ranksrgan: Super resolution generative adversarial networks with learning to rank." IEEE Transactions on Pattern Analysis and Machine Intelligence 44.10 (2021): 7149-7166.
> >
> > [11] Blau, Yochai, and Tomer Michaeli. "The perception-distortion tradeoff." Proceedings of the IEEE conference on computer vision and pattern recognition. 2018.

---

> ### Author Response · Authors · 2023-11-21
> **Response to Reviewer MvoB (part 3/3)**
>
> **Q4: The authors utilized 5 or 6 metrics to evaluate different methods, but did not discuss the results carefully. The numbers in Table 5 can not clearly validate the effectiveness of the proposed FFL and AFF-Net.**
>
> A4: Thanks for the comment. To thoroughly evaluate our methods, we have included 6 metrics in the paper: PSNR, SSIM, LPIPS, FID, MUSIQ, and NIQE. Among these metrics, PSNR and SSIM are objective quality metrics that measure the similarity of pixels or local areas between two images. However, they may not always align with human perception of image quality and often contradicts subjective quality metrics[1]. This study primarily focuses on perception-based super-resolution and prioritizes subjective metrics. LPIPS and FID are reference-based metrics that assess perceptual quality. LPIPS specifically evaluates perceptual consistency at the image level, while FID measures the overall distributional differences between real and generated images. Therefore, FID captures the overall style difference between two image folders rather than image-level similarity. On the other hand, MUSIQ and NIQE are non-reference metrics that evaluate image quality. MUSIQ evaluates image quality with multi-scale input, while NIQE assesses the naturalness of the image by analyzing statistical features. As shown in Table 5, we obtained reasonable values for all metrics. For example, AFF+FFL outperforms the baseline in terms of SSIM, LPIPS, MUSIQ, and NIQE. Considering the popularity of LPIPS and NIQE, we will discuss these two metrics in detail in the following part.
>
>
> **Quantitative comparison.** In the following Tab.1, both AFF-Net and FFL Loss show improvements in these metrics compared to the baseline. When both are used together, there is a 7.3% improvement in LPIPS and a 22.1% improvement in NIQE compared to the baseline. In the following Tab.2, comparing UNet and AFF, they have the same architecture except that AFF utilizes convolution operations in the frequency domain. The better LPIPS and nearly comparable NIQE scores indicate the effectiveness of frequency operations in FCD.
>
> Table1 Ablation studies of FFL Loss and AFF-Net
> | FFL Loss  | AFF-Net | LPIPS$\downarrow$  | NIQE$\downarrow$ |
> | --------  | --------| ------ | ------ |
> |           |         | 0.3038 | 5.6093 |
> |  &#10004; |         | 0.2892 | 4.5789 |
> |    |  &#10004;      | 0.2808 | 4.4337 |
> |  &#10004; |&#10004; | 0.2815 | 4.3675 |
>
> Table2 Comparsion AFF-Net and UNet
> |  Model  | LPIPS$\downarrow$  | NIQE$\downarrow$ |
> | --------| ------ | -------- |
> |  UNet+FFL| 0.2856 | 4.3652 |
> |  FCD+FFL | 0.2815 | 4.3675 |
>
> **Qualitative results and spectrum analysis.** We also add qualitative results and spectrum analysis in Appendix C.2. The [qualitative results](https://drive.google.com/file/d/1dvaUvtknGEjVPnD7OiDQVRLqS0PSwr1V/view?usp=sharing) illustrate that FCD effectively compensates for high-frequency details, adding realistic features such as animal fur and skin texture, thereby avoiding oversmooth. Additionally, the spectrum analysis presented in [Fig.10](https://drive.google.com/file/d/15ECHoCPny0Xx_wHPI2LLLsyxEAhGjmK3/view?usp=sharing) demonstrates that FCD generates images with a **higher correlation coefficient** between their spectrum and the spectrum of the ground truth, indicating improved frequency consistency.
>
> **Application to other tasks.** We have **expanded the application of FCD to include image reconstruction and text-to-image generation**. The corresponding details are provided in Appendix D, and the results are showcased in [Fig.11](https://drive.google.com/file/d/1TWHiYD4XEvnmCQ8VkwIZXxwFifA6mfrB/view?usp=sharing) and [Fig.12](https://drive.google.com/file/d/1kLk0RGZrYVNcPiil7B02dXgKOtevPALK/view?usp=sharing). Notably, the completion of high-frequency details in human faces enhances the realism of the generated images, demonstrating the effectiveness of FCD.
>
> **Reference**
>
>
> [1] Blau, Yochai, and Tomer Michaeli. "The perception-distortion tradeoff." Proceedings of the IEEE conference on computer vision and pattern recognition. 2018.

---

### Official Review · Reviewer_dphS · 2023-10-30

**Soundness:** 3 good
**Presentation:** 3 good
**Contribution:** 3 good
**Rating:** 6
**Confidence:** 4

**Summary:**

This paper proposes Sampling-Space MoE to enlarge the diffusion model without necessitating a substantial increase in training and inference resources. To address the issue of information loss caused by the latent representation of the diffusion model, the author further presents a frequency-compensated decoder to refine the details of super-resolution images. Experimental results on both Blind and Non-Blind SR datasets demonstrate that the proposed method obtain good performance.

**Strengths:**

1. Appealing visual results are obtained with the proposed method.
2. The paper is well-written and organized, making it easy to understand the proposed framework and its contributions.

**Weaknesses:**

1. Ablation studies lack qualitative results, and the results in Table 5 do not offer strong evidence for the effectiveness of the proposed FCD.
2.The visual results are sometimes good but with severe hallucination, and quantitative results are not always good on LPIPS and NIQE (These two metrics are generally more convincing in real image restoration).
3. Since the compared methods including GAN-based methods and Diffuison-based method, it is insufficient to evaluate these methods only based on restoration metrics. The author should also make a comparison on model complexity, inference speed, and GPU usage.

**Questions:**

see the Weaknesses part.

---

> ### Author Response · Authors · 2023-11-21
> **Response to Reviewer dphS (part 1/2)**
>
> We thank the reviewer for the constructive comments. We have incorporated your comments into the revision. We added qualitative results of ablation studies in Appendix C.1 and Appendix C.2. We would appreciate it if you have any further questions or comments.
>
> **Q1: Ablation studies lack qualitative results**
> A1: Thanks for your advice. We added qualitative results for SS-MoE and FCD in [Appendix C.1](https://drive.google.com/file/d/1H6VX1IvApOhvYul1QYR9mmCMpRW7GRox/view?usp=sharing) and [Appendix C.2](https://drive.google.com/file/d/1dvaUvtknGEjVPnD7OiDQVRLqS0PSwr1V/view?usp=sharing). The results demonstrate the effectiveness of both SS-MoE and FCD. By utilizing Sample MoE and Space MoE, we are able to obtain sharp high-resolution images with high fidelity. Furthermore, FCD compensates for more high-frequency details, resulting in the generation of realistic textures.
>
> **Q2: The results in Table 5 do not offer strong evidence for the effectiveness of the proposed FCD.**
> A2:
> 1. FCD's quantitative results on LPIPS and NIQE(the two more convincing metrics as you mentioned) show its effectiveness. In the following Tab.1, both AFF-Net and FFL Loss show improvements in these metrics compared to the baseline. When both are used together, there is a 7.3% improvement in LPIPS and a 22.1% improvement in NIQE compared to the baseline. In the following Tab.2, comparing UNet and AFF, they have the same architecture except that AFF utilizes convolution operations in the frequency domain. The better LPIPS and nearly comparable NIQE scores indicate the effectiveness of frequency operations in FCD.
>
> Table1 Ablation studies of FFL Loss and AFF-Net
> | FFL Loss  | AFF-Net | LPIPS$\downarrow$  | NIQE$\downarrow$ |
> | --------  | --------| ------ | ------ |
> |           |         | 0.3038 | 5.6093 |
> |  &#10004; |         | 0.2892 | 4.5789 |
> |    |  &#10004;      | 0.2808 | 4.4337 |
> |  &#10004; |&#10004; | 0.2815 | 4.3675 |
>
> Table2 Comparsion AFF-Net and UNet
> |  Model  | LPIPS$\downarrow$  | NIQE$\downarrow$ |
> | --------| ------ | -------- |
> |  UNet+FFL| 0.2856 | 4.3652 |
> |  FCD+FFL | 0.2815 | 4.3675 |
>
> 2. As depicted in [Appendix C.2](https://drive.google.com/file/d/1dvaUvtknGEjVPnD7OiDQVRLqS0PSwr1V/view?usp=sharing), the qualitative results illustrate that FCD effectively compensates for high-frequency details, adding realistic features such as animal fur and skin texture, thereby avoiding oversmooth. Additionally, the spectrum analysis presented in [Fig.10](https://drive.google.com/file/d/15ECHoCPny0Xx_wHPI2LLLsyxEAhGjmK3/view?usp=sharing) demonstrates that FCD generates images with a higher correlation coefficient between their spectrum and the spectrum of the ground truth, indicating improved frequency consistency.
>
> 3. We have expanded the application of FCD to include image reconstruction and text-to-image generation. The corresponding details are provided in Appendix D, and the results are showcased in [Fig.11](https://drive.google.com/file/d/1TWHiYD4XEvnmCQ8VkwIZXxwFifA6mfrB/view?usp=sharing) and [Fig.12](https://drive.google.com/file/d/1kLk0RGZrYVNcPiil7B02dXgKOtevPALK/view?usp=sharing). Notably, the completion of high-frequency details in human faces enhances the realism of the generated images, demonstrating the effectiveness of FCD.
>
> **Q3: The visual results are sometimes good but with severe hallucination**
> A3: Super-resolution is an ill-posed task where a degraded image can have multiple high-definition image solutions through compensation and restoration, resulting in a *one-to-many* possibility. Diffusion models, known for their strong generative capabilities, excel at filling in missing image details thus resulting in more realistic images. However, this enhanced generative ability sometimes can introduce hallucinations, where the predicted high-resolution images may differ from the ground truth but still appear reasonable. **It is worth noting that hallucinations are not exclusive to diffusion models; they are a general problem in super-resolution.** Even existing common GAN-based methods, like BSRGAN and Real-ESRGAN can also exhibit hallucinations, such as the [example](https://drive.google.com/file/d/1WDUB0mD6xnm4mJu-Mc3TIYmPrtylB9g2/view?usp=drive_link).

---

> ### Author Response · Authors · 2023-11-21
> **Response to Reviewer dphS (part 2/2)**
>
> **Q4: Quantitative results are not always good on LPIPS and NIQE (These two metrics are generally more convincing in real image restoration).**
> A4: We have achieved state-of-the-art results in terms of LPIPS and NIQE on the DIV2K dataset for 4x Blind SR. Furthermore, our method has outperformed existing diffusion-based approaches on RealSR and DRealSR. For 8x Non-Blind SR, we have achieved SOTA results in terms of LPIPS, and our NIQE score ranks second among all methods.
>
> Indeed, our method does not achieve SOTA performance on all tasks (Blind and Non-Blind SR) and datasets. It is challenging to simultaneously address multiple super-resolution tasks with the same network architecture and data degradation pipelines. The limitation of the model's generalization is also mentioned in the paper. Exploring better image degradation pipelines may improve this aspect, but it is beyond the scope of this study.
>
> **Q5: Since the compared methods including GAN-based methods and Diffuison-based method, it is insufficient to evaluate these methods only based on restoration metrics. The author should also make a comparison on model complexity, inference speed, and GPU usage.**
> A5: Thanks for your suggestion. We list parameter number, TFLops, inference time and GPU usage in the following table. The tests were conducted on a V100 GPU using the Torch 2.0 environment. In diffusion-based methods, we use 200 timesteps. It is true that diffusion-based methods require more computational resources compared to GAN-based methods. This is why we propose MoE to efficiently scale diffusion-based super-resolution. It is important to note that **diffusion-based methods have a stronger ability to generate realistic details but come with a higher computational cost**. On the other hand, **GAN-based methods are lightweight but may have limitations in generating fine details**. These are the characteristics of the two families of methods.
>
> Furthermore, with the help of SS-MoE, our method only experiences a 6.1% increase in FLOPs compared to LDM. The increase in inference time can be attributed to the impact of GPU load on inference speed, while the increased GPU memory is a result of the cache of sampling experts. Addressing these issues can be achieved through engineering optimization strategies, which are beyond the scope of this paper. **Compared to stablesr, our method requires fewer TFLops and inference time while achieving better LPIPS and NIQE scores**, as shown in Tab.1 of our paper. This demonstrates the effectiveness of our proposed method.
>
> | Method | Num of Params | TFLops | Inference Time | GPU usage |
> | -------- | -------- | -------- |--------| --------|
> |  BSRGAN  | 16.70       | 0.2937 |0.0413 s | 1432M   |
> |  Real-ESRGAN+ | 16.70  | 0.2937 |0.0413 s | 1432M   |
> |  FeMaSR       | 33.48  | 0.3761 |0.0755 s | 1904M   |
> |  LDM          | 168.95 | 33.43  |5.5693 s | 4778M   |
> |  StableSR     | 1409.11| 86.27  |16.1110 s | 8966M  |
> |  Ours         | 605.30 | 35.47  |7.4702 s | 9160M   |

---

### Author Response · Authors · 2023-11-21
**Summary of Revision**

We would like to thank the reviewers for their valuable time and effort invested in reviewing our work. We genuinely appreciate the constructive feedback, and we are encouraged by the recognition of the novelty and the contribution of our work. In response to the feedback, we provide individual responses to address each reviewer’s concerns, and an updated manuscript.

**Reviewer dphS** primarily expressed concerns about the qualitative results of the ablation studies and efficiency analysis. In response, we have included more qualitative results of both SS-MoE and FCD, and included an analysis of computation cost, inference speed and GPU usage for GAN-based and diffusion-based methods.

**Reviewer MvoB** questioned our contribution and the performance of our model. In response, we have clarified our contributions in two aspects and analyzed the performance via quantitative results, qualitative results and user study in Appendix B and Appendix C .

**Reviewer krWG** raised concerns about the effectiveness of FCD. We acknowledge some missing analysis, and we have added quantitative results, spectrum analysis and qualitative results of FCD in Appendix C to show its function

**Reviewer uAFu** questioned the motivation of SS-MoE and FCD and the performance of our model compared with prior studies. In response, we have clarified our idea and validated it via experiments. We have analyzed the performance our model achieves and demonstrated its competitiveness.

---

> ### Author Response · Authors · 2023-11-22
> **Looking forward to reviewers' feedbacks**
>
> Dear Reviewers,
>
> We would like to express our gratitude for your dedicated review and constructive suggestions. As the deadline for revision and discussion is approaching (**on November 22, at 23:59 UTC-12h**), we kindly request your further valuable feedback. Your re-evaluation of our work and any additional insightful comments would be greatly appreciated. We sincerely hope to continue the discussion with you and address any remaining concerns.
>
> Thank you once again for your time and consideration.
>
> Best regards,
>
> The authors

---

### Meta-Review · Area_Chair_1M3q · 2023-12-09

**Metareview:**

This paper proposes to solve single-image super-resolution problem with latent diffusion model. To compensate for the compression error in the latent space, frequency augmentation network is introduced. To relieve the increased computational cost from the use of diffusion model, SS-MoE structure.

There were several shared concerns from the reviewers.
- limited novelty in frequency compensation module as it is common in handling the loss from autoencoders and in SS-MoE as MoE approach is very simple
- unclear motivation of SS-MoE
- inconsistent quality improvement compared with SoTA methods

In the rebuttal, the authors claimed that
- The frequency compensation module is first used for the proposed latent diffusion model problem
- SS-Moe is novel and the motivation is described in the paper.
- LPIPS and NIQE are more important than other quality metrics.

Sampling MoE is adopted from the previous works ((Xue et al., 2023; Feng et al., 2022; Balaji et al., 2022). For Space-MoE, the benefits are described but the motivation is not very clear from the paper.

LPIPS and NIQE are generally considered valid perceptual metrics, however, it is common that perception-distortion trade-off should be considered in image restoration literature. For example, from a single neural network architecture, it is possible to draw a model with better LPIPS and NIQE scores by just adjusting the optimization goals. If the superiority in terms of LPIPS and NIQE are to be claimed, more detailed proofs should be presented.
The authors provided a user study in the Appendix, however, a better explanation of evaluation metrics should be provided for clarity. In the main body of the paper, there is no description of the user study. While the user study and visual examples provide clue for the improved perceptual quality, the exposition has room to improve.

**Justification For Why Not Higher Score:**

There were major concerns pointed out from the reviewers. I looked into the paper details and the author rebuttal, however, the authors' responses are not very convincing. The concerns are partially addressed, however, without significance.

**Justification For Why Not Lower Score:**

N/A

---

### Decision · Program_Chairs · 2024-01-16

Reject